# Unlocking Long-Term Dependencies in Spiking Neural Networks with a Recurrent LIF Memory Module

## Abstract

Processing long sequence data such as speech requires models to maintain long-term dependencies, which is challenging for recurrent spiking neural networks (RSNNs) due to high temporal dynamics in neuron models that leak stored information in their membrane potentials, and face vanishing gradients during backpropagation through time. These issues can be mitigated by employing more complex neuron designs, such as ALIF and TC-LIF, but these neuron-level solutions often incur high computational costs and complicate hardware implementation, undermining the efficiency advantages of SNNs. Here we propose an architectural-level solution that leverages the dynamical interactions of a few LIF neurons to enhance long-term information storage. The memory capability of this LIF-based micro-circuit is adaptively modulated by global recurrent connections of the RSNN, contributing to selective enhancement of temporal information retention, and ensures stable gradient gain when propagation through time. The proposed model outperforms previous methods including LSTM, ALIF, and TC-LIF in long sequence tasks, achieving 96.52% accuracy on the PS-MNIST dataset. Furthermore, our method also provides a compelling efficiency advantage, yielding up to 277× improvement compared to conventional models such as LSTM. This work paves the way for building cost-effective, hardware-friendly, and interpretable spiking neural networks for long sequence modeling.

## 1 Introduction

Spiking neural networks (SNNs) offer energy-efficient computing paradigms by leveraging brain-inspired neuron models as activation functions to enable sparse and event-driven computations (Roy et al., 2019). The leaky integrate-and-fire (LIF) is the most widely adopted neuron model in SNNs, which integrates input signals and generates a spike once the membrane potential exceeds its firing threshold (Gerstner & Kistler, 2002). To enhance temporal resolution for sequential inputs, the LIF incorporates a leak mechanism that effectively filters out irrelevant long-term information, making SNNs a good candidate for temporal signal processing tasks using recurrent spiking neural networks (RSNNs) architectures (Bellec et al., 2018).

Nonetheless, the performance of LIF-based RSNNs, particularly in long-sequence modeling, still faces three major challenges: (1) the leak mechanism, while beneficial for short-term dynamics, causes the LIF neuron to forget earlier inputs, hindering the capture of long-term dependencies; (2) RSNNs with simple recurrent connections lack adaptive mechanisms to dynamically regulate information flow based on input salience, making them ineffective at distinguishing useful information from noises; (3) training RSNNs via backpropagation through time (BPTT) (Werbos, 2002) is impeded by the gradient vanishing problem, which greatly limits the model's overall performance.

To overcome the short-term memory limits of the vanilla LIF neuron, several complex neuron models have been proposed to incorporate additional mechanisms such as adaptive thresholds (Yin et al., 2021), compartmental dynamics (Zhang et al., 2024a) , or variable time constants (Fang et al., 2021a) in individual spiking neurons. Although these approaches have demonstrated improved robustness in long-sequence modeling, their model complexity leads to high computational cost and additional design overhead for neuromorphic hardware.

Rather than relying on the intrinsic properties of individual neurons for long-term memory, an alternative approach is to leverage the collective dynamics at the architectural level. For example, long short-term memory (LSTM) networks in artificial neural networks (ANNs) address the long-sequence problem by introducing a gated cell state. However, the gating mechanism is not natively supported by most neuromorphic processors, since many neuromorphic processors, such as Loihi 2 (Abreu et al., 2025), rely on specialized arithmetic logic units (ALUs) to emulate neuron operations which only support low-precision, element-wise operations such as integer and fixed-point addition, multiplication, and bitwise shifts, which imposes inherent limitations when approximating nonlinear functions such as tanh, sigmoid or exponential functions. Directly performing these non-native operations using these ALUs might suffer significant accuracy loss and require support from external high precision units. Alternatively, RSNNs suitable for long-sequence tasks may be obtained through complex network structure designs Shen et al. (2023), but have not been validated to comprehensively support selective long-term memory and robust training in RSNNs. Spiking LMUFormer (Liu et al., 2024), an attention-based SNN, was proposed for long sequence modeling task, but involves complex architecture and Conv1d operations, hindering its efficient deployment in neuromorphic hardware.

In this work, we propose a novel recurrent architecture of RSNN for long sequence modeling, based on hierarchical recurrent connections including a compact local LIF recurrent memory module (LRMM). The LRMM used four vanilla LIF neurons to form input, output and the memory loop, dynamically regulating the stored information without gating units, while offering native compatibility with neuromorphic hardware. The local memory loop also enhances gradient propagation under BPTT with less vanishing gradient, demonstrating stable gradient retention for training of RSNNs. We evaluated our model on several long-sequence benchmarks, including PS-MNIST, SHD, SSC, and the Binary Adding task. Our approach outperforms standard LIF networks and complex neuron-centric models such as TC-LIF in terms of accuracy, gradient stability, and robustness to long sequences. It also demonstrated performance comparable to complex architectures such as LSTM and Spiking LMUFormer, while using only a fraction of parameters (0.15M for LRMM vs 1.61M for Spiking LMUFormer on PS-MNIST) and maintaining excellent computing efficiencies with $26\times$ less energy than Spiking LMUFormer and $277\times$ to that of LSTM. Our contributions are summarized as follows:

- We design a vanilla LIF based recurrent memory module that incorporates local memory loop for long-term information retention without complex neuron designs.
- We employ a hierarchical recurrent architecture that combines global recurrent connections and local recurrent memory to dynamically and selectively regulate the memory of input data.
- We show that the memory loop improves gradient propagation under BPTT and enhances the gradient retention factor, thereby mitigating the vanishing gradient problem in training.
- We validate our model on four long-sequence benchmarks (PS-MNIST, SHD, SSC, and Binary Adding task), demonstrating improved accuracy, stable training dynamics, and superior energy efficiency.

## 2 RELATED WORK

**Long-Term Memory in SNNs.** A key challenge in SNNs is retaining information over a long time. Several neuron-centric approaches address this issue by modifying LIF dynamics, such as adaptive thresholds in ALIF (Bellec et al., 2018), radial dynamics in RadLIF (Bohnstingl et al., 2022), and dual-compartment coupling in TC-LIF (Yin et al., 2023). Although effective, they increase model complexity and require hardware-specific tuning, limiting their efficiency and scalability.

**Gated Recurrent Models.** Recurrent architectures such as LSTM (Hochreiter & Schmidhuber, 1997) and GRU (Cho et al., 2014) achieve strong performance in sequential tasks by explicitly gating and storing information over time. However, both conventional and spiking counterparts, such as Spiking-LSTM (Lotfi Rezaabad & Vishwanath, 2020), rely on complex gating and expensive state updates, which limit their suitability for neuromorphic computing.

**Structural Complexity in SNNs** Another line of work enhances temporal processing in SNNs by introducing architectural complexity, such as locally recurrent motifs (Zhang et al., 2024c), small-

world connectivity (Pan et al., 2024), and brain-inspired topologies (Wang et al., 2024). These studies suggest that structural complexity can benefit temporal modeling in SNNs, yet they do not provide explicit mechanisms to sustain long-term dependencies. Spiking LMUFormer (Liu et al., 2024) introduces long-range memory through state-space models, but its deep convolutional architecture leads to high computational and parameter costs, making it less suitable for neuromorphic hardware.

## 3 METHOD

### 3.1 VANILLA LIF BASED RECURRENT MEMORY MODULE

**Spiking Neuron Model.** We employ the vanilla LIF neuron as the fundamental computational unit in our RSNNs. The membrane potential $u(t)$ evolves over time according to the following differential equation:

$$\tau \frac{du(t)}{dt} = -(u(t) - u_{reset}) + RI(t) \tag{1}$$

Here, $\tau$ is the membrane time constant, $R$ is the resistance, $I(t)$ is the synaptic input, and $u_{\text{reset}}$ is the reset potential. A spike $S(t)$ is emitted when the membrane potential exceeds the threshold $V_{\text{th}}$, after which it is reset to $u_{\text{reset}}$. For practical implementation, we discretize the equation using the Euler method. Assuming $u_{reset} = 0$ and $R = \tau$, the discrete-time update with soft reset is:

$$u[t+1] \; = \; \left(1 - \frac{1}{\tau}\right)(u[t] - V_{th}\,S[t]) \; + \; I[t], \qquad S[t] \; = \; \Theta(u[t] - V_{th}). \tag{2}$$

Here, $\Theta(\cdot)$ is the Heaviside step function, which outputs 1 when its argument is positive and 0 otherwise. This prevents runaway spiking while preserving the residual subthreshold voltage, and the leak factor $(1 - \frac{1}{\tau})$ still governs the temporal decay between spikes.

**Recurrent Memory Module Design Based on LIF Neurons.** We propose a lightweight and interpretable recurrent memory module composed entirely of vanilla LIF neurons, as shown in Figure 1. Each module contains four LIF units with distinct functional roles, i.e., input integration neuron $N_\text{I}$ that processes incoming and global feedback signals, memory neuron $N_\text{M}$ that maintains long-term temporal memory, context $N_\text{C}$ that combines the memory and input, and output control neuron $N_\text{O}$ determines the readout information. The LRMM module is plugged in the global recurrent architecture of RSNN, which receives three modulatory currents, i.e., $I_\text{I}, I_\text{F}, I_\text{O}$ that dynamically regulate the information flow based on input salience. The interaction through structured global and local connections, enabling the LRMM module to retain long-term information within a fully spike-driven and biologically plausible framework.

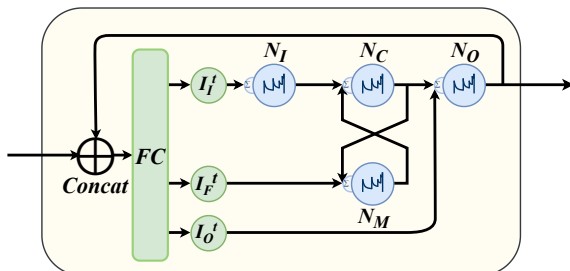

Figure 1: Vanilla LIF based recurrent memory module framework.

Each of the three input currents $j \in \{\text{I}, \text{F}, \text{O}\}$ is computed from the same combination of the current input $\text{input}[t]$ and the previous output spike $S_{N_O}[t-1]$, using separate fully connected layers.

$$I_j[t] \; = \; \Phi\Big(\boldsymbol{W}_j\,[\,\text{input}[t]; S_{N_O}[t-1]\,] + \boldsymbol{b}_j\Big), \quad j \in \{\text{I}, \text{F}, \text{O}\}, \tag{3}$$

where the modulation function $\Phi$ interpolates between the standard sigmoid and a piecewise-linear (PL) hard-sigmoid:

$$\Phi(z) = (1-m)\,\sigma(z) \; + \; m\,\text{PL}(z), \qquad m \in [0,1], \tag{4}$$

$$\text{PL}(z) = \text{clip}\Big(0.5 + \frac{z}{2a},\, 0,\, 1\Big), \qquad a > 0, \tag{5}$$

with $\text{clip}(x, 0, 1) = \min(\max(x, 0), 1)$. During training, we anneal $m_t$ from 0 to 1. At inference, we set $m = 1$ so that $I_j[t] = \text{PL}(\cdot)$, which provides a hardware-friendly approximation.

The inputs of four neurons in the LRMM module can be formulated as:

$$I_{N_I}[t] = k_I \cdot I_I[t] \tag{6}$$

$$I_{N_M}[t] = w_{C,M} \cdot S_{N_C}[t-1] + k_F \cdot I_F[t] \tag{7}$$

$$I_{N_C}[t] = w_{I,C} \cdot S_{N_I}[t] + w_{M,C} \cdot S_{N_M}[t] \tag{8}$$

$$I_{N_O}[t] = w_{C,O} \cdot S_{N_C}[t] + k_O \cdot I_O[t] \tag{9}$$

where $S_X[t]$ denotes the spike output of neuron $X$ at time $t$ and $I_X[t]$ denotes its corresponding input current and $k_X$ means the trainable scaling factor. The weight $w_{a,b}$ specifies the synaptic connection from neuron $a$ to neuron $b$. All parameters $\{\boldsymbol{W}_j, \boldsymbol{b}_j\}_{j \in \{I,F,O\}}$ in eq. (3) and all synaptic scalars $w_{a,b}$ in eqs. (6) to (9) are learnable and time-shared across $t$. This schedule ensures causal updates and avoid algebraic loops, since $N_M$ depends on $S_{N_C}[t-1]$ while $N_C$ consumes the freshly produced $S_{N_I}[t]$ and $S_{N_M}[t]$.

The proposed LRMM enables effective temporal modeling via exploiting dynamic interactions among event-driven LIF neurons. Unlike approaches that rely on complex neuron-level modifications, our method provides an architectural level solution that supports both short-term modulation and long-term memory integration within a fully spike-based architecture. The module introduces adaptive regulation of input salience, allowing it to selectively determine which information is integrated, stored, and read out. The use of vanilla LIF and compact RSNN architecture enhances runtime efficiency and compatibility with neuromorphic hardware implementations.

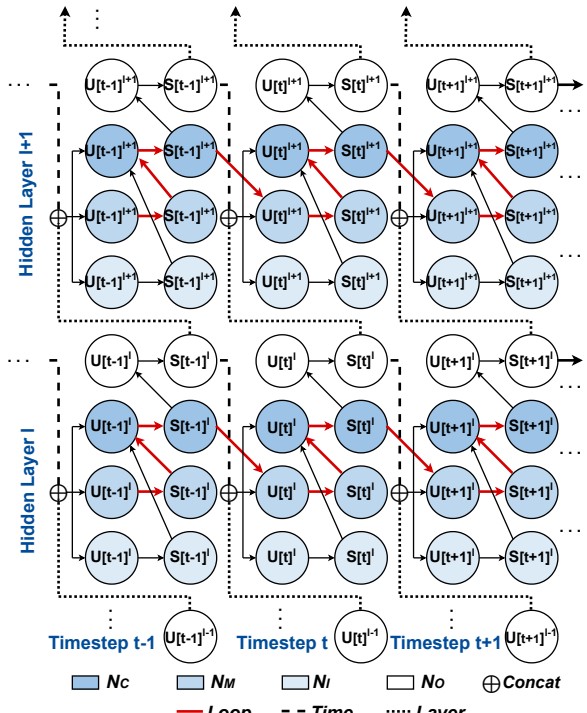

Figure 2: Forward information flow in the proposed recurrent memory module. Red bold arrows indicate temporal feed-forward paths between $N_M$ and $N_C$. Dashed arrows indicate temporal and layer-wise forward connections. Note that during BPTT, gradients propagate in reverse direction of the arrows.

## 3.2 STABILITY ANALYSIS OF BACKPROPAGATION THROUGH TIME (BPTT)

In this subsection, we analyze the BPTT dynamics using the surrogate computation graph during training. The *temporal* gradient component from membrane leakage and reset is decoupled from the *spatial* gradient component of reverse-mode accumulation along spike-to-current pathways. The separated pathways make the sources of gradient amplification and attenuation explicit, enabling a principled stability analysis. For clarity, Figure 2 illustrates the forward information flow in the proposed recurrent memory module. We further investigate how local recurrent loops contribute to stable long-range gradient propagation across extended temporal horizons.

**Notation.** For clarity we defer all symbols and training-time surrogate details to Appendix A.2. In the main text we only use $A_X[t]$ as the one-step temporal Jacobian defined in equation 10.

$$A_X[t] = \frac{\partial U_X[t+1]}{\partial U_X[t]} = \alpha_X \Big( 1 - V_{\text{th},X} \, \sigma'_X(U_X[t] - V_{\text{th},X}) \Big). \tag{10}$$

where $A_X[t]$ is the one-step temporal gain of neuron $X$. It measures how membrane leak and threshold reset contract or expand the mapping along time, with $A_X[t] \approx \alpha_X$ when $\sigma'_X \approx 0$ far from threshold and a smaller value near threshold due to the subtractive reset.

**Per-neuron one-step temporal recursion.** At time $t$, $U_X[t+1]$ depends on $U_X[t]$ (leak+reset) and exogenous $I_X[t+1]$; $S_X[t]$ depends on $U_X[t]$. We keep forward firing hard and use a smooth surrogate only in back propagation:

$$S_X[t] = \sigma_X(U_X[t] - V_{\text{th},X}). \tag{11}$$

Using the soft-reset update and equation 10, the one-step temporal gain is $A_X[t]$ as defined above. With the boxcar surrogate A.2 and $H_w = V_{\text{th},X}/2$ (normalize $V_{\text{th},X} = 1$), $\sigma'_X(\cdot) \in \{1, 0\}$ and thus

$$A_X[t] \in \{\alpha_X, 0\}, \quad A_X[t] = \alpha_X \text{ if } |U_X[t] - V_{\text{th},X}| > H_w, \text{ and } A_X[t] = 0 \text{ otherwise.} \tag{12}$$

By the chain rule, the loss gradient to $U_X[t]$ decomposes into a temporal path and a spike path:

$$gU_X[t] = \underbrace{A_X[t]\, gU_X[t+1]}_{\text{temporal}} + \underbrace{\sigma'_X(U_X[t] - V_{\text{th},X})\, gS_X[t]}_{\text{spatial}}. \tag{13}$$

**Per-neuron one-step BPTT recursions.** By equation 13, with $\sigma'_X[t] \equiv \sigma'_X(U_X[t] - V_{\text{th},X})$ and $A_X[t]$ from equation 10, the adjoint for each neuron satisfies. Details are in A.2:

$$gU_{N_I}[t] = A_{N_I}[t]\, gU_{N_I}[t+1] + \sigma'_{N_I}[t]\, w_{I,C}\, gU_{N_C}[t], \tag{14}$$

$$gU_{N_M}[t] = A_{N_M}[t]\, gU_{N_M}[t+1] + \sigma'_{N_M}[t]\, w_{M,C}\, gU_{N_C}[t], \tag{15}$$

$$gU_{N_C}[t] = A_{N_C}[t]\, gU_{N_C}[t+1] + \sigma'_{N_C}[t]\Big(w_{C,O}\, gU_{N_O}[t] + w_{C,M}\, gU_{N_M}[t+1]\Big). \tag{16}$$

**Loop-Induced Effective Temporal Gain in $N_M$–$N_C$.** We analyze how the $N_M$–$N_C$ loop affects the one-step temporal operator in BPTT. Starting from the per-unit adjoint recursions, we define the loop couplings $\beta_t$ and $\gamma_t$ and apply a single substitution. From the per-neuron BPTT recursions of $N_C$ and $N_M$ in equation 15 and equation 16, we define

$$\beta_t := \sigma'_{N_C}[t]\, w_{C,M}, \qquad \gamma_t := \sigma'_{N_M}[t]\, w_{M,C}. \tag{17}$$

Substituting $gU_{N_M}^{t+1} = A_{N_M}^{t+1} gU_{N_M}^{t+2} + \gamma_{t+1} gU_{N_C}^{t+1}$ into $gU_{N_C}^t$ yields

$$gU_{N_C}^t = \underbrace{(A_{N_C}^t + \beta_t \gamma_{t+1})}_{\text{effective temporal gain}} gU_{N_C}^{t+1} + \beta_t A_{N_M}^{t+1} gU_{N_M}^{t+2} + \sigma'_{N_C}[t]\, w_{C,O}\, gU_{N_O}^t. \tag{18}$$

$$gU_{N_M}^t = \underbrace{(A_{N_M}^t + \gamma_t \beta_t)}_{\text{effective temporal gain}} gU_{N_M}^{t+1} + \gamma_t A_{N_C}^t gU_{N_C}^{t+1} + \gamma_t\, \sigma'_{N_C}[t]\, w_{C,O}\, gU_{N_O}^t. \tag{19}$$

Equations equation 18 and equation 19 yield direct one-step recursions $gU_{N_C}^t \to gU_{N_C}^{t+1}$ and $gU_{N_M}^t \to gU_{N_M}^{t+1}$ with effective temporal gains $A_{N_C}^t + \beta_t \gamma_{t+1}$ and $A_{N_M}^t + \gamma_t \beta_t$, respectively. Compared with the leak-only LIF baseline where the gains equal $A_{N_C}^t$ and $A_{N_M}^t$, the loop contributes an additional coupling term $\beta_t \gamma_{t+1}$ or $\gamma_t \beta_t$, which establishes a direct pass-through across consecutive steps and reduces reliance on the leak factor $A_X^t$.

Recall $A_X[t] = \alpha_X(1 - \sigma'_X[t])$ from equation 10, and define

$$G_{N_C}^t := A_{N_C}^t + \beta_t \gamma_{t+1} = \alpha_{N_C}(1 - \sigma'_{N_C}[t]) + \sigma'_{N_C}[t]\, \sigma'_{N_M}[t+1]\, w_{C,M}\, w_{M,C}, \tag{20}$$

$$G_{N_M}^t := A_{N_M}^t + \gamma_t \beta_t = \alpha_{N_M}(1 - \sigma'_{N_M}[t]) + \sigma'_{N_M}[t]\, \sigma'_{N_C}[t]\, w_{M,C}\, w_{C,M}. \tag{21}$$

The quantities $G_{N_C}^t$ and $G_{N_M}^t$ comprise two complementary components that are active in different operating regimes. For $G_{N_C}^t$ one has

$$G_{N_C}^t = \underbrace{\alpha_{N_C}(1 - \sigma'_{N_C}[t])}_{\text{off-threshold contribution}} + \underbrace{\sigma'_{N_C}[t]\, \sigma'_{N_M}[t+1]\, w_{C,M}\, w_{M,C}}_{\text{near-threshold loop contribution}},$$

and for $G_{N_M}^t$ one has an analogous decomposition. Thus each gain contains an off-threshold term proportional to $1 - \sigma'$ and a near-threshold term proportional to $\sigma'$. This complementary structure substantially improves the ability of gradients to propagate over time, since at least one component remains active across typical operating regions. In particular, when $\sigma'_{N_C}[t] \approx 1$ and $\sigma'_{N_M}[t+1] \approx 1$ or when $\sigma'_{N_M}[t] \approx 1$ and $\sigma'_{N_C}[t] \approx 1$, the loop contribution dominates and gradients are conveyed through the interconnecting synapses with magnitude controlled by $w_{C,M}\, w_{M,C}$. This reduces sequences of zero temporal gain and preserves gradient connectivity at spike-adjacent time steps.

| Datasets | Method | Recurrent | Vanilla LIF | Parameters | Accuracy (%) |
|---|---|---|---|---|---|
| PS-MNIST (T=784) | LIFZhang et al. (2024b) | Y | Y | 0.155M | 80.39 |
| | LSTM (Rusch & Mishra, 2021) | Y | N | 0.27M | 92.90 |
| | GLIF (Yao et al., 2022) | Y | N | 0.15M | 90.47 |
| | ALIF (Yin et al., 2021) | Y | N | 0.15M | 94.30 |
| | BRFN (Higuchi et al., 2024) | N | N | 0.068M | 95.20 |
| | TC-LIF (Zhang et al., 2024a) | Y | N | 0.063M/0.15M | 92.69 / 95.36 |
| | Spiking LMUFormer (Liu et al., 2024) | Y | N | 1.61M | 97.92 |
| | **LRMM (ours)** | Y | Y | 0.15M | **96.52** |
| | **LRMM-ALIF (ours)** | Y | N | 0.15M | **97.39** |
| SSC (T=100) | LIF (Cramer et al., 2020) | Y | N | 0.11M | 50.90 |
| | TC-LIF (Zhang et al., 2024a) | Y | N | 0.11M | 61.90 |
| | LSTM (Cramer et al., 2020) | Y | N | 0.43M | 73.10 |
| | SNN-CNN (Sadovsky et al., 2023) | N | N | N/A | 72.03 |
| | ALIF (Yin et al., 2021) | Y | N | N/A | 74.20 |
| | SpikGRU (Dampfhoffer et al., 2022) | Y | N | 0.28M | 77.00 |
| | RadLIF (Bittar & Garner, 2022) | N | N | 3.9M | 77.40 |
| | Spiking LMUFormer (Liu et al., 2024) | Y | N | 2.13M | 78.58 |
| | **LRMM (ours)** | Y | Y | 0.20M | **79.75** |
| | **LRMM-ALIF (ours)** | Y | N | 0.20M | **80.51** |
| SHD (T=100) | LIF (Cramer et al., 2020) | Y | N | 0.108M | 71.40 |
| | LSTM (Cramer et al., 2020) | Y | N | 0.43M | 89.20 |
| | TC-LIF (Zhang et al., 2024a) | Y | N | 0.15M | 88.91 |
| | ALIF (Yin et al., 2021) | Y | N | N/A | 90.40 |
| | RadLIF (Bittar & Garner, 2022) | Y | N | 3.9M | 94.62 |
| | Spiking LMUFormer (Liu et al., 2024) | Y | N | 2.12M | 91.04 |
| | **LRMM (ours)** | Y | Y | 0.42M | **94.70** |
| | **LRMM-ALIF (ours)** | Y | N | 0.42M | **95.32** |
| Binary Adding (T=100) | LIF (Ma et al., 2025) | N | Y | 0.04M | 53.35 |
| | PLIF (Fang et al., 2021b) | Y | N | N/A | 53.25 |
| | adLIF (Bellec et al., 2018) | Y | N | N/A | 68.00 |
| | ALIF (Yin et al., 2021) | Y | N | N/A | 99.05 |
| | GLIF (Teeter et al., 2018) | Y | N | N/A | 63.60 |
| | TC-LIF (Zhang et al., 2024a) | Y | N | N/A | 19.90 |
| | LM-H (Hao et al., 2023) | Y | N | N/A | 96.10 |
| | CLIF (Huang et al., 2024) | Y | N | N/A | 64.30 |
| | DH-LIF (Zheng et al., 2024) | Y | N | N/A | 99.35 |
| | **LRMM (ours)** | Y | Y | 0.15M | **99.55** |
| | **LRMM-ALIF (ours)** | Y | N | 0.15M | **100.00** |

Table 1: **Results on Temporal Benchmarks.** Evaluation on PS-MNIST, SSC, SHD, and Binary Adding benchmarks, reporting accuracy (%) and parameter counts. LRMM employs only vanilla LIF neurons, while **LRMM-ALIF** replaces $N_I$ and $N_O$ with ALIF neurons. Across all tasks, LRMM achieves competitive or even state-of-the-art accuracy while using significantly fewer parameters.

## 4 EXPERIMENTS

### 4.1 EXPERIMENTAL SETUP

**Datasets.** We evaluate our models on four widely used benchmark datasets for sequential and event-driven learning, i.e., SHD (Cramer et al., 2020), SSC (Warden, 2018), PSMNIST (Le et al., 2015), and Binary Adding (Ma et al., 2025). These datasets are chosen to cover a diverse range of temporal modeling challenges, including event-based auditory processing, speech command recognition, long-range dependency reasoning, and numerical sequence addition. A detailed description of each dataset is provided in Appendix A.3.

**Models.** Sequential inputs are fed directly into the network without spike encoding. All recurrent computations are handled by our proposed *LIF-based recurrent memory module (LRMM)*, which uses structured recurrent connections among LIF neurons to support memory formation in a fully spike-driven and biologically plausible manner. This recurrent design enables stable gradient propagation, long-term temporal integration, and selective retention of salient input patterns. By default, we use a two-layer LRMM backbone with 128 units per layer, followed by a linear classifier on the final hidden state.

**Training Details.** All training configurations, including hyperparameters and optimization strategies, are provided in Appendix A.4.

**Baseline Models and Comparative Methodology.** Detailed baseline configurations and comparison settings are provided in Appendix A.5.

## 4.2 MAIN RESULTS

We present a comprehensive evaluation of LRMM to demonstrate its effectiveness across four dimensions, i.e., accuracy, gradient behavior, memory capability, and gating selectivity in long horizon temporal classification. **First**, on standard benchmarks including PS-MNIST, SHD, SSC, and Binary Adding, LRMM achieves strong and competitive accuracy with comparable or reduced parameter counts under a unified training protocol. **Second**, on long-range dependency settings, LRMM maintains accurate retrieval despite extended delays and noise, demonstrating robust long-term memory. **Third**, analysis of circuit-level activity shows that the recurrent loop adaptively modulates memory traces, enhancing informative segments and suppressing irrelevant ones. Causal interventions further demonstrate the necessity of this adaptive recurrence for long-term retention.

**Fourth**, analysis of gradient flow shows that LRMM mitigates vanishing gradients and preserves more temporal credit assignment over long horizons, as quantified by the Gradient Retention Factor.

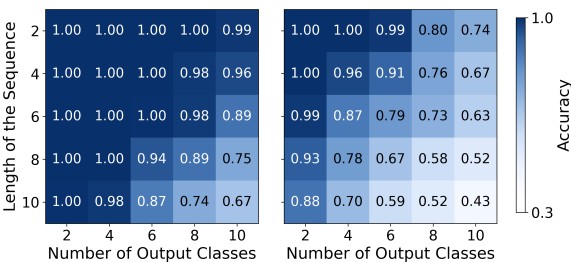

**Results on Temporal Benchmarks.** Under a unified protocol with matched parameter budgets, LRMM attains strong accuracy across four long sequence benchmarks. On PS-MNIST, LRMM achieves 96.52% with 0.15M parameters, exceeding TC-LIF at 95.36% with 0.15M. On SSC, LRMM reaches 79.75% with

Figure 3: The accuracy of copy task(T=40). Ours 2 layers 128 dim network (left) compare with 2 layers 1024 dim network ALIF (right).

0.20M, outperforming RadLIF at 77.40% with 3.9M. On SHD, LRMM obtains 94.70% with 0.42M compared with 94.62% for RadLIF with 3.9M. On Binary Adding, LRMM records 99.55% with 0.15M, slightly higher than the 99.35% baseline. Compared to Spiking LMUFormer, a deeper architecture that incorporates both attention and state space memory mechanisms, our LRMM achieves strong performance with a significantly smaller parameter budget. While Spiking LMUFormer reaches higher accuracy on PS-MNIST, our LRMM-ALIF achieves a comparable 97.39% accuracy while using only 0.15M parameters, compared to 1.61M in Spiking LMUFormer. On SSC and SHD, LRMM outperforms Spiking LMUFormer by a margin of 1.17% and 3.66% respectively, using over six times fewer parameters. These results show that our simple recurrent LIF-based circuit matches or outperforms deeper and more complex models on long sequence tasks. These results, summarized in Table 1, indicate consistent improvements at compact model sizes.

Our framework supports heterogeneous neuron integration to further enhance performance. By replacing the input encoder $N_I$ and readout neuron $N_O$ with ALIF units, while maintaining the memory neuron $N_M$ and aggregation neuron $N_C$ as LIF, our method achieve higher accuracy of 97.39% on PSMNIST, 80.51% on SSC, 95.32% on SHD, and 100.00% on Binary Adding under the same parameter counts.

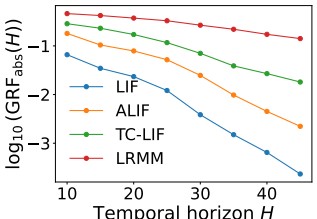

**Evaluating Long-Term Memory.** We evaluate the long-term memory capacity of LRMM using the copy task (Graves et al., 2014; Bellec et al., 2020), a canonical benchmark for measuring temporal credit assignment across extended delays. Each input consists of a sequence of $L \in \{2, \ldots, 10\}$ tokens drawn from an alphabet of size $K \in \{2, \ldots, 10\}$, followed by a stop signal and a fixed delay of $delay_t = 20$ time steps. After receiving the readout cue, the model must reproduce the original sequence in exact order and length. Figure 3 reports test

Figure 4: Log-scale absolute GRF $\log_{10}(\text{GRF}_{\text{abs}}(H))$ across temporal horizons $H$. MALC consistently outperforms LIF, ALIF, and TC-LIF, especially as $H$ increases.

accuracy across the $(L, K)$ grid. With two layers of 128 circuits, LRMM maintains near-perfect accuracy ($\geq 0.99$) on short sequences across all alphabet sizes. Under the most challenging configuration ($L=10, K=10$), LRMM achieves 66.9% accuracy, outperforming a large ALIF model with two layers and 1024 neurons, which reaches only 43.0%. In a moderately difficult setting ($L=8, K=8$), LRMM attains 88.9%, while the ALIF baseline reaches 58.4%. Despite using significantly fewer neurons and parameters, LRMM consistently outperforms ALIF across all conditions. We attribute this advantage to the memory loop between $N_M$ and $N_C$, which effectively preserves task-relevant information during the long delay intervals.

**Adaptive Recurrent Dynamics.** To analyze how LRMM adaptively processes information across memory stages, we use a modified copy task which contains structured noise and explicit control signals, as shown in Figure 5(a). Each input sequence spans 20 time steps, structured as two cycles of alternating informative segments and noise intervals (5 steps each). At each time step, a control signal indicates whether the current input should be remembered or ignored.

After the entire input sequence, a readout signal is issued, and the model must output the concatenated informative segments in order. As shown in Figures 5(b)–(d) and (g), during noise segments, both the forget signal $F$ and input signal $I$ are significantly reduced compared to informative data, indicating that the model avoids forgetting stored content while ignoring irrelevant input. At the same time, Figures 5(e), (f), and (h) show stronger Hamming correlation between $N_M$ and $N_C$, suggesting more stable internal recurrence. In contrast, during informative data, both $F$ and $I$ increase, reflecting active integration of new input with existing memory, accompanied by more dynamic activity between $N_M$ and $N_C$. During the readout phase, the output $O$ becomes selectively active, not merely propagating memory but enabling targeted information retrieval.

These results suggest that LRMM achieves robust memory control through adaptive recurrent mechanisms that filter, store, and extract information in noisy temporal settings.

**Gradient Retention Analysis.** We further evaluate the temporal gradient stability by computing the relative Gradient Retention Factor (GRF) A.6.1 across training. As shown in As shown in Figure 6, LRMM achieves consistently higher single-step geometric gain compared to the baseline without inter-loop recurrence, exceeding it by more than $1.5\times$ on average. This indicates significantly improved gradient flow and more

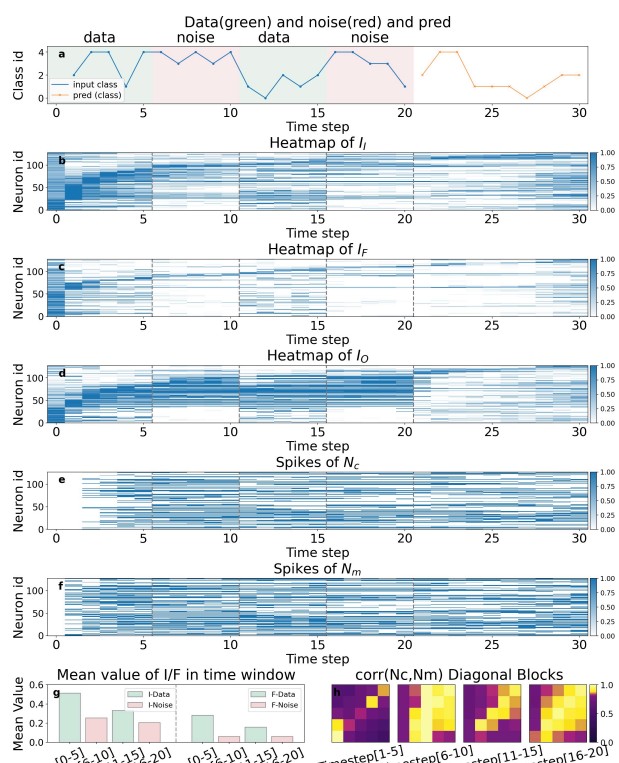

Figure 5: The input sequence consists of 20 time steps, as shown in (a). The green regions denote informative symbols that must be remembered, while the red regions represent noise. (b–d) illustrate the temporal dynamics of the I, F, and O gates across all neurons, while their respective mean values are summarized in (g). The I and F gates exhibit high activation during informative segments and remain nearly inactive during noise, whereas the O gate is selectively activated during the readout phase. (e) and (f) show the spiking activity of the $N_C$ and $N_M$ neurons, respectively. The correlation matrix between them is shown in (h).

effective temporal credit assignment over long horizons. As shown in Figure 4, LRMM achieves the highest absolute GRF A.6.1 across all tested horizons. The gap becomes increasingly significant as $H$ grows, indicating that the structured recurrent feedback in LRMM enables more stable gradient propagation over long sequences, compared to LIF, ALIF and TC-LIF.

| Model | Theoretical energy | Energy (nJ) |
|-------|-------------------|-------------|
| LRMM | $(7n\,E_{\mathrm{MAC}} + n(3m\,\mathrm{Fr}_{\mathrm{in}} + 3n\,\mathrm{Fr}_{N_O}/4 + \mathrm{Fr}_{N_I} + \mathrm{Fr}_{N_M} + 2\,\mathrm{Fr}_{N_C})\,E_{\mathrm{AC}})/4$ | 76.11 |
| TC-LIF | $2n\,E_{\mathrm{MAC}} + (mn\,\mathrm{Fr}_{\mathrm{in}} + (n^2+2n)\,\mathrm{Fr}_{\mathrm{out}})\,E_{\mathrm{AC}}$ | 212.27 |
| LIF | $n\,E_{\mathrm{MAC}} + (mn\,\mathrm{Fr}_{\mathrm{in}} + (n^2+n)\,\mathrm{Fr}_{\mathrm{out}})\,E_{\mathrm{AC}}$ | 186.60 |
| Spiking LMUFormer | $mn\,E_{\mathrm{MAC}} + S(10n^2 + nK)E_{\mathrm{AC}}.$ | 2004 |
| LSTM | $(4(mn+n^2) + 17n)\,E_{\mathrm{MAC}}$ | 21145 |

Table 3: Energy consumption comparison on SHD. 2 layers with 512 neurons.

## 4.3 ABLATION STUDY

To evaluate the contribution of each structural component in our LIF memory circuit, we conduct a series of controlled ablation experiments. All variants are trained under the same protocol on PS-MNIST and SHD. We measure classification accuracy and BPTT gradient stability to assess the effect of circuit modifications. Specifically, we ablate: (1) the recurrent feedback from the memory neuron $N_C$ to the controller $N_M$, (2) the recurrent output path from $N_O$ to the current layer, and (3) the gating mechanism, replacing all three gates with a shared static input gate.

As shown in Table 2, all three ablations cause consistent performance drops across both datasets, validating the necessity of feedback modulation, temporal recurrence, and gate specialization in the proposed memory circuit.

**Memory-State Modulated Feedback.** Removing the feedback from the memory neuron $N_C$ to the controller $N_M$ results in the most significant degradation: PS-MNIST accuracy drops from 96.52% to 86.11%, and SHD drops from 94.70% to 86.28%. This ablation breaks the memory-control loop, impairing the circuit's ability to retain and coordinate long-term information. **Recurrent Memory Path.** Eliminating the output recurrence from $N_O$ to the

| Dataset | Ablation Setting | Accuracy(%) ↑ |
|---------|------------------|---------------|
| PS-MNIST | Full Model | 96.52 |
|  | w/o $N_C \rightarrow N_M$ connection | $86.11_{\downarrow 9.41}$ |
|  | w/o Recurrent connections | $93.74_{\downarrow 2.78}$ |
|  | w/o Gate Separation | $92.35_{\downarrow 4.17}$ |
| SHD | Full Model | 94.70 |
|  | w/o $N_C \rightarrow N_M$ connection | $86.28_{\downarrow 8.42}$ |
|  | w/o Recurrent connections | $92.47_{\downarrow 2.23}$ |
|  | w/o Gate Separation | $90.81_{\downarrow 3.89}$ |

Table 2: Ablation study of LRMM.

current layer weakens temporal integration. Accuracy drops moderately by 2.78% on PS-MNIST and 2.23% on SHD. Although the model retains basic temporal processing via internal delays, the lack of global recurrence leads to reduced gradient stability and more localized memory formation, especially in longer sequences.

**Gate Separation.** Replacing the three distinct gates ($I_I$, $I_F$, $I_O$) with a single shared input gate impairs selective signal routing. This simplification causes accuracy to drop by 4.17% on PS-MNIST and 3.89% on SHD, suggesting that dedicated gating enables fine-grained temporal filtering of relevant versus irrelevant information streams.

## 4.4 HIGH ENERGY EFFICIENCY

We use the accounting: total energy = #MAC·$E_{\mathrm{MAC}}$ + #AC·$E_{\mathrm{AC}}$. $E_{\mathrm{AC}}$=0.9 pJ, $E_{\mathrm{MAC}}$=4.6 pJ. (Horowitz, 2014) Setup for SHD: 2 hidden layers, $\mathrm{hiddenneurons} = 512, \mathrm{n} = 20$, SHD firing rate:0.114. LRMM Firing rates:$\mathrm{Fr}_{N_I}$=(0.168, 0.196), $\mathrm{Fr}_{N_O}$=(0.311, 0.269), $\mathrm{Fr}_{N_C}$=(0.366, 0.330), $\mathrm{Fr}_{N_M}$=(0.274, 0.197), $\mathrm{Fr}_{out}$ = 0.08; LIF Firing rates: (0.274,0.226), $\mathrm{Fr}_{out}$=0.085; TC-LIF firing rates: (0.294, 0.241), $\mathrm{Fr}_{\mathrm{out}}$=0.108.

**Energy efficiency.** On SHD, LRMM attains the lowest dynamic energy among all recurrent baselines, as shown in Table 3. Thanks to its localized memory loops and sparsely activated recurrent connections, spike activity is concentrated within a small subset of synapses, which substantially reduces the number of event-driven accumulation operations compared to more densely activated LIF/TC-LIF layers and gate-based LSTMs. Empirically, this translates to a measured energy of

76.11 nJ per step for LRMM, compared to 186.60 nJ for LIF, 212.27 nJ for TC-LIF and 2004 nJ for Spiking LMUFormer, i.e., about $2.45\times$, $2.79\times$ and $26.33\times$ lower energy, respectively. LSTM is even more expensive, consuming 21145 nJ per step, so LRMM is roughly $277\times$ more energy-efficient. Overall, LRMM trades a small linear increase in MACs for a much larger reduction in spike-driven accumulation events, yielding the best energy–performance balance among all compared models on SHD.

## 5 SUMMARY AND DISCUSSION

We introduced the LIF Recurrent Memory Module (LRMM), a lightweight spiking memory architecture built entirely from vanilla LIF neurons with fixed, structured connectivity. By augmenting LIF dynamics with a localized memory loop, LRMM achieves long-range temporal integration while preserving low firing rates, low parameter count, and high energy efficiency. The architecture ensures stable gradient flow, enabling effective temporal credit assignment without relying on trainable adaptation or explicit synaptic delays. Extensive experiments on benchmark sequence tasks demonstrate that LRMM achieves high performance among SNNs, while consuming up to $59\%$ less event-driven energy than LIF and over $277\times$ fewer energy than LSTM. Despite these advantages, LRMM has not yet been deployed on neuromorphic hardware. Future directions include hardware-aligned implementations, multi-scale memory integration, and scaling to large real-world environments such as continuous control and autonomous agents.

## 6 ETHICAL CONSIDERATIONS AND COMPLIANCE WITH THE OPEN SCIENCE POLICY

### 6.1 ETHICAL CONSIDERATIONS

This study proposes the LRMM architecture to improve the long-term memory capacity of spiking neural networks using structured recurrence and vanilla LIF neurons. All experiments were conducted using publicly available benchmark datasets such as PS-MNIST, SHD, and SSC. The work does not involve the use of personal data, content generation, or any human-related applications. Our research is intended for theoretical analysis and academic benchmarking, with a focus on advancing the understanding of memory mechanisms in energy-efficient spiking models.

### 6.2 COMPLIANCE WITH THE OPEN SCIENCE POLICY

To support reproducibility and transparency, we provide all necessary details for reproducing our experiments in the appendix. This includes a comprehensive description of the datasets used, the evaluation metrics such as Gradient Retention Factor, and the experimental configurations. We also include an extended explanation of how LRMM enhances BPTT gradient stability. An anonymized code repository is referenced in the appendix to allow reviewers to verify our implementation and results without compromising the double-blind review process.

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

## A APPENDIX

### A.1 AUTHOR DISCLOSURE OF LLM USAGE

In accordance with the ICLR 2026 policy on the use of large language models (LLMs), we disclose that LLMs were used solely for language polishing purposes during the preparation of this manuscript. All LLM-generated content was manually reviewed and edited by the authors to ensure accuracy and appropriateness. LLMs were not used for literature review, method design, experiment implementation, analysis, or any other substantive aspect of the research.

## A.2 BPTT PROOF

**Notation.** We index time $t$ and units $X \in \{N_\mathrm{I}, N_\mathrm{M}, N_\mathrm{C}, N_\mathrm{O}\}$ and current gates $j \in \{\mathrm{I}, \mathrm{F}, \mathrm{O}\}$. Each unit keeps membrane $U_X[t]$, spike $S_X[t]$, input $I_X[t]$, leak $\alpha_X \in (0, 1)$, threshold $V_{\mathrm{th}, X}$. We reuse $gZ[t] = \partial\mathcal{L}/\partial Z[t]$ for any symbol $Z$ (e.g., $gU_X[t], gS_X[t], gI_j[t]$) and parameter gradients $\partial\mathcal{L}/\partial w$.

**Soft-reset update and derivation of $A_X[t]$.** During training we adopt a soft-reset LIF update

$$U_X[t+1] = \alpha_X(U_X[t] - V_{\mathrm{th}, X} S_X[t]) + I_X[t+1], \tag{22}$$

and treat $I_X[t+1]$ as exogenous when differentiating w.r.t. $U_X[t]$ so that inter-step structural effects are accounted for separately in the spatial graph. With the surrogate $S_X[t] = \sigma_X(U_X[t] - V_{\mathrm{th}, X})$ and $\partial S_X[t]/\partial U_X[t] = \sigma_X'(\cdot)$, we obtain

$$A_X[t] = \frac{\partial U_X[t+1]}{\partial U_X[t]} = \alpha_X\Big(1 - V_{\mathrm{th}, X}\, \sigma_X'(U_X[t] - V_{\mathrm{th}, X})\Big). \tag{23}$$

This coefficient quantifies the local temporal gain induced jointly by membrane leak and subtract-threshold reset.

**Surrogate gradients (training only).** We keep the forward dynamics hard and invoke surrogates only in the backward pass. For the modulation clamp in equation 3, when the forward path uses the piecewise-linear hard-sigmoid $\mathrm{PL}(z)$, its derivative is approximated by the logistic-sigmoid derivative:

$$\frac{\partial I_j[t]}{\partial z_j[t]} \approx \sigma(z_j[t])(1 - \sigma(z_j[t])), \qquad z_j[t] = W_j\,[\,\mathrm{input}[t]; S_{N_O}[t-1]\,] + b_j, \tag{24}$$

where $\sigma(z) = 1/(1 + e^{-z})$. For spikes in equation 2, we adopt the rectangular surrogate with half-width $H_w > 0$:

$$\frac{\partial S_X[t]}{\partial U_X[t]} = \begin{cases} \dfrac{1}{2H_w}, & |U_X[t] - V_{\mathrm{th}, X}| \le H_w, \\ 0, & \text{otherwise.} \end{cases} \tag{25}$$

Gradients through the reset factor $(1 - S_X[t])$ use the same spike surrogate $\partial S_X[t]/\partial U_X[t]$. At inference time we employ the hard $\mathrm{PL}(\cdot)$ clamp and the Heaviside firing function without surrogates.

**Step-by-step BPTT for each neuron in details.** We write $\sigma_X'[t] \equiv \sigma_X'(U_X[t] - V_{\mathrm{th}, X})$ and use $A_X[t]$ from equation 10. Input neuron $N_I$:

$$gU_{N_I}[t] = \underbrace{A_{N_I}[t]\, gU_{N_I}[t+1]}_{\text{temporal}} + \sigma_{N_I}'[t]\, \underbrace{w_{I,C}\, gU_{N_C}[t]}_{\text{spatial}} \tag{26}$$

Neuron $N_M$:

$$gU_{N_M}[t] = \underbrace{A_{N_M}[t]\, gU_{N_M}[t+1]}_{\text{temporal}} + \sigma_{N_M}'[t]\, \underbrace{w_{M,C}\, gU_{N_C}[t]}_{\text{spatial}} \tag{27}$$

Neuron $N_C$:

$$gU_{N_C}[t] = \underbrace{A_{N_C}[t]\, gU_{N_C}[t+1]}_{\text{temporal}} + \sigma_{N_C}'[t](\underbrace{w_{C,O}\, gU_{N_O}[t]}_{\text{spatial}} + \underbrace{w_{C,M}\, gU_{N_M}[t+1]}_{\text{temporal}}) \tag{28}$$

Neuron $N_O$:

$$gU_{N_O}[t] = \underbrace{\Big(A_{N_O}[t] + \sigma_{N_O}'[t]\, c_O[t+1]w_{N_O,O}k_O\Big)\, gU_{N_O}[t+1]}_{\text{temporal}}$$

$$+ \sigma_{N_O}'[t]\Big(\underbrace{c_I[t+1]w_{N_O,I}k_I\, gU_{N_I}[t+1]}_{\text{temporal}} \tag{29}$$

$$+ \underbrace{c_F[t+1]w_{N_O,F}k_F\, gU_{N_M}[t+1]}_{\text{temporal}} + \underbrace{\frac{\partial\mathcal{L}}{\partial S_{N_O}[t]}}_{\text{spatial}}\Big)$$

Here $c_j[t+1]$ collects the local slope along the path $S_{N_O}[t] \to I_j[t+1]$ through the modulation $\Phi$ and the corresponding linear map, for $j \in \{\mathrm{I}, \mathrm{F}, \mathrm{O}\}$.

### A.3 DATASETS

**SHD (Spiking Heidelberg Digits)** (Cramer et al., 2020) is a neuromorphic dataset that consists of spike-based representations of spoken digits (0–9), recorded using a model of the auditory periphery. Each sample is represented as a sequence of spatio-temporal spike events across 700 input channels over a duration of 1 second. The dataset contains 8,144 training samples and 2,264 test samples. It is particularly suited for evaluating the temporal processing capabilities of spiking neural networks (SNNs).

**SSC (Spiking Speech Commands)** (Warden, 2018) is an event-driven version of the Google Speech Commands dataset, converted into spike trains using biologically inspired auditory models. It includes 35 spoken keywords mapped to 20 classes, with a total of 8,000 training and 2,000 test samples. Like SHD, SSC emphasizes temporal precision and robustness in spike-based representations, making it a suitable testbed for SNN-based models.

**PSMNIST (Permuted Sequential MNIST)** (Le et al., 2015) is a sequential version of the standard MNIST handwritten digit dataset. Each $28 \times 28$ image is flattened into a 784-dimensional sequence, and then a fixed random permutation is applied to the sequence order. The dataset contains 60,000 training and 10,000 test samples. PSMNIST is widely used to benchmark recurrent and sequential models due to its requirement for long-range dependency modeling.

**Binary Adding (long-range marked-sum).** Following Ma et al. (2025), this synthetic sequence task is designed to evaluate a model's ability to capture long-range temporal dependencies. Each input contains two binary sequences of length $T$: a value sequence $x_1 \in \{0,1\}^T$ and a marker sequence $x_2 \in \{0,1\}^T$. The marker $x_2$ selects 9 positions within $x_1$, and the label is the sum of $x_1$ at these positions, yielding a 10-class target (0–9). The model must process the entire sequence before prediction, making it a strict test of temporal integration. We generate 50,000 training and 2,000 test samples, and vary $T$ to control task difficulty.

### A.4 TRAINING DETAILS.

All LRMM units share the same LIF parameters: a trainable leak factor initialized to 0.95, a fixed firing threshold $V_{\mathrm{th}} = 1.0$, and a reset potential $V_{\mathrm{reset}} = 0$. All feedforward and recurrent weights are initialized using Xavier uniform initialization. We adopt a boxcar surrogate gradient with width $w = 1.0$. Full input sequences are used without truncation during BPTT. Training is performed using the Adam optimizer with $\beta_1 = 0.9$ and $\beta_2 = 0.999$, an initial learning rate of $1 \times 10^{-2}$, a batch size of 128, and a total of 50 epochs. Classification is performed based on the firing rates of the output neurons, and the model is trained using the standard cross-entropy loss.

### A.5 BASELINE MODELS AND COMPARATIVE METHODOLOGY.

We evaluate three categories of baselines under controlled model capacity and training settings. First, to test whether network-level structure can replace neuron-level complexity, we compare against ALIF (Yin et al., 2021), TC-LIF (Zhang et al., 2024a), RadLIF (Bittar & Garner, 2022), and DCLS-Delays (Hammouamri et al., 2023), which incorporate adaptive thresholds, compartmental dynamics, radial memory, or delay-based recurrence. Second, we include an LSTM (Hochreiter & Schmidhuber, 1997) with matched parameter budget to assess compute cost and energy efficiency relative to a standard ANN baseline. Third, we test a stacked LIF network without the LRMM loop, isolating the effect of our proposed structural design. These comparisons help disentangle neuron-intrinsic mechanisms from architectural complexity, and quantify the impact of LRMM under consistent experimental conditions.

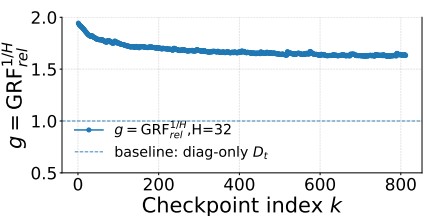

Figure 6: Single-step geometric gain $g = (\mathrm{GRF}_{\mathrm{rel}})^{1/H}$ on SHD ($H = 32$). LRMM shows consistently higher gain than the baseline without inter-loop connections.

Regarding training configurations, ALIF, RadLIF, TC-LIF use normalization layers. For surrogate gradients, ALIF uses a triangular surrogate, RadLIF uses a Gaussian surrogate, TC-LIF uses a triangular surrogate.

### A.6 METRICS

#### A.6.1 METRICS FOR TEMPORAL STABILITY AND GRADIENT PROPAGATION

We use a compact metric suite to characterize (i) global multi-step gradient retention and (ii) the local stability of the MALC microcircuit.

**Absolute Gradient Retention Factor (GRF).** Let $J_t = \partial u_{t+1}/\partial u_t$ be the per-step Jacobian of a neuron/microcircuit state (scalar for LIF, coupled blocks for ALIF/TC-LIF, and a $3\times3$ operator for MALC). Over horizon $H$ starting at $t$,

$$\text{GRF}_{\text{abs}}(H;t) = \Big\| \prod_{k=0}^{H-1} J_{t+k} \Big\|_2, \quad \text{GRF}_{\text{abs}}(H) = \text{median}_{t \in \mathcal{T}} \, \text{GRF}_{\text{abs}}(H;t).$$

Larger values indicate stronger long-range gradient preservation.

**Loop-Sensitive Relative GRF (MALC).** To isolate the $N_M \leftrightarrow N_C$ loop effect, define the MALC temporal operator

$$M_t = \begin{bmatrix} A_{N_M}^t + \gamma_t \beta_t & \gamma_t A_{N_C}^t \\ \beta_t & A_{N_C}^t \end{bmatrix}, \quad D_t = \text{diag}(A_{N_M}^t, A_{N_C}^t),$$

with $A_X^t = \alpha_X(1 - V_{\text{th},X}\,\sigma'_X(U_X^t - V_{\text{th},X}))$, $\beta_t = \sigma'_{N_C}(U_{N_C}^t - V_{\text{th},N_C})\,w_{C,M}$, $\gamma_t = \sigma'_{N_M}(U_{N_M}^t - V_{\text{th},N_M})\,w_{M,C}$, where $\sigma'$ is the surrogate derivative. The loop contribution is

$$\text{GRF}_{\text{rel}}(H;t) = \frac{\| \prod_{k=0}^{H-1} M_{t+k} \|_2}{\| \prod_{k=0}^{H-1} D_{t+k} \|_2}, \quad \text{GRF}_{\text{rel}}(H) = \text{median}_{t \in \mathcal{T}} \, \text{GRF}_{\text{rel}}(H;t).$$

Values $> 1$ indicate loop-induced amplification beyond diagonal decay.

**Spectral Radius.** Local stability is summarized by $\rho(M_t) = \max_i |\lambda_i(M_t)|$: $\rho(M_t) < 1$ suggests contraction, $\rho(M_t) \approx 1$ near-critical memory, and $\rho(M_t) > 1$ potential instability.

### A.7 ENERGY COUNTING OF LRMM UNDER THE EVENT-DRIVEN CONVENTION

**Counting convention.** We follow the *event-driven* convention used in prior SNN energy tables: (i) event-triggered computations (including projections, gates, and loop updates) are accounted as accumulations with unit cost $E_{\text{AC}}$; (ii) only the per-neuron temporal state update (leakage/reset) is treated as a dense multiply–accumulate (MAC), yielding $n E_{\text{MAC}}$ per step.

**Notation.** Let $m$ be the input width and $n$ the total number of neurons in the layer. Denote by $Fr_{\text{in}}$ the average input firing ratio, and by $Fr_{N_O}$ the average output firing ratio (neuron $N_O$). Within the local loop, $Fr_{N_I}$, $Fr_{N_M}$, and $Fr_{N_C}$ denote the firing ratios associated with nodes $N_I$, $N_M$, and $N_C$, respectively. The symbols $E_{\text{MAC}}$ and $E_{\text{AC}}$ denote the unit energy of a MAC and an accumulation, respectively. Event-triggered operations are charged as $E_{\text{AC}}$; temporal leakage/reset contributes the $n E_{\text{MAC}}$ baseline.

**Operation counting per step.** We decompose the per-step energy into four parts and then aggregate.

**(A) Per-neuron temporal state update.** Each neuron incurs one dense update per step,

$$E_A = n\, E_{\text{MAC}}. \tag{30}$$

**(B) Input-driven accumulations.** The three gates consume input events over $m \times \frac{n}{4}$ connections,

$$E_B = \frac{3}{4} mn\, Fr_{\text{in}}\, E_{\text{AC}}. \tag{31}$$

**(C) Output-driven accumulations.** Output spikes at node $N_O$ fan out to all $n/4$ local modules along three paths per module,

$$E_C = \frac{3}{16} n^2 \, Fr_{N_O} \, E_{\text{AC}}.$$ (32)

**(D) Two-node loop accumulations.** Within each module, the loop $(N_M \leftrightarrow N_C)$ and other connections add local event interactions. Aggregating over all modules yields,

$$E_D = \frac{1}{4} n (Fr_{N_I} + Fr_{N_M} + 2 \, Fr_{N_C}) \, E_{\text{AC}}.$$ (33)

**(E) Input current scaling.** The three input currents $I_I$, $I_F$, and $I_O$ each receive a learned scalar weight,

$$E_E = \frac{3}{4} n \, E_{\text{MAC}}.$$ (34)

**Aggregate cost.** Summing equation 30–equation 34 gives

$$E_{\text{LRMM/step}} = \frac{7}{4} n \, E_{\text{MAC}} + \left( \frac{3mn}{4} Fr_{\text{in}} + \frac{3n^2}{16} Fr_{N_O} + \frac{n}{4} (Fr_{N_I} + Fr_{N_M} + 2 Fr_{N_C}) \right) E_{\text{AC}}.$$ (35)

A.8 ENERGY COUNTING FOR TC-LIF, LIF, LSTM, AND SPIKING LMUFORMER ON SHD

**Setup.** Two hidden layers with total neurons per layer $n = 512$. Layer-1 takes external input of width $m_1 = 700$. Layer-2 takes the output of Layer-1 with effective width $m_2 = 512$. The theoretical per-step energies used are

$$E^{(\ell)}_{\text{TC-LIF}} = 2n \, E_{\text{MAC}} + (m_\ell n \, Fr^{(\ell)}_{\text{in}} + (n^2 + 2n) \, Fr_{\text{out}}) \, E_{\text{AC}},$$ (36)

$$E^{(\ell)}_{\text{LIF}} = n \, E_{\text{MAC}} + (m_\ell n \, Fr^{(\ell)}_{\text{in}} + (n^2 + n) \, Fr_{\text{out}}) \, E_{\text{AC}},$$ (37)

$$E^{(\ell)}_{\text{LSTM}} = (4(m_\ell n + n^2) + 17n) \, E_{\text{MAC}}.$$ (38)

**Firing rates.** LIF: $(Fr^{(1)}_{\text{in}}, Fr^{(2)}_{\text{in}}) = (0.274, 0.226)$, $Fr_{\text{out}} = 0.085$.
TC-LIF: $(Fr^{(1)}_{\text{in}}, Fr^{(2)}_{\text{in}}) = (0.294, 0.241)$, $Fr_{\text{out}} = 0.108$.

LIF

Layer-1:
$$E^{(1)}_{\text{LIF}} = 512 \, E_{\text{MAC}} + (700 \cdot 512 \cdot 0.274 + (512^2 + 512) \cdot 0.085) \, E_{\text{AC}}$$
$$= 512 \, E_{\text{MAC}} + 120527.36 \, E_{\text{AC}}.$$ (39)

Layer-2:
$$E^{(2)}_{\text{LIF}} = 512 \, E_{\text{MAC}} + (512 \cdot 512 \cdot 0.226 + (512^2 + 512) \cdot 0.085) \, E_{\text{AC}}$$
$$= 512 \, E_{\text{MAC}} + 81570.304 \, E_{\text{AC}}.$$ (40)

Two-layer total:
$$E_{\text{LIF,total}} = 1024 \, E_{\text{MAC}} + 202097.664 \, E_{\text{AC}} = 186.60 \text{nJ}.$$ (41)

TC-LIF

Layer-1:
$$E^{(1)}_{\text{TC-LIF}} = 1024 \, E_{\text{MAC}} + (700 \cdot 512 \cdot 0.294 + (512^2 + 2 \cdot 512) \cdot 0.108) \, E_{\text{AC}}$$
$$= 1024 \, E_{\text{MAC}} + 133791.744 \, E_{\text{AC}}.$$ (42)

Layer-2:
$$E^{(2)}_{\text{TC-LIF}} = 1024 \, E_{\text{MAC}} + (512 \cdot 512 \cdot 0.241 + (512^2 + 2 \cdot 512) \cdot 0.108) \, E_{\text{AC}}$$
$$= 1024 \, E_{\text{MAC}} + 91598.848 \, E_{\text{AC}}.$$ (43)

Two-layer total:
$$E_{\text{TC-LIF,total}} = 2048 \, E_{\text{MAC}} + 225390.592 \, E_{\text{AC}} = 212.27 \text{nJ}.$$ (44)

LSTM

Layer-1:

$$E_{\text{LSTM}}^{(1)} = (4(700 \cdot 512 + 512^2) + 17 \cdot 512) \, E_{\text{MAC}}$$
$$= 2490880 \, E_{\text{MAC}}. \tag{45}$$

Layer-2:

$$E_{\text{LSTM}}^{(2)} = (4(512 \cdot 512 + 512^2) + 17 \cdot 512) \, E_{\text{MAC}}$$
$$= 2105856 \, E_{\text{MAC}}. \tag{46}$$

Two-layer total:

$$E_{\text{LSTM,total}} = 4596736 \, E_{\text{MAC}} = 21145 \text{nJ}. \tag{47}$$

SPIKING LMUFORMER

**Energy of Spiking LMUFormer on SHD (1 LMU block, $d = 512$).** We estimate the per–timestep compute energy as in the Spiking LMUFormer paper (Liu et al., 2024), with $E_{\text{MAC}} = 4.6$ pJ, $E_{\text{AC}} = 0.9$ pJ, and average firing rate $S = 0.15$. For SHD, the network contains a 5-layer convolutional patch embedding, one spiking LMU block ($d = 512$), a two-layer convolutional channel mixer, and a $512 \to 20$ fully-connected classifier, together with batch-normalization (BN) and spiking-neuron (SN) layers.

**Operation counts.** We first count synaptic operations for each MAC/AC-bearing component:

$$\text{Conv patch (1st layer, dense MAC):} \quad N_{\text{conv0}} = 700 \times 512,$$
$$\text{Conv patch (4 sparse Conv1d layers):} \quad N_{\text{patch}} = 4 \times 512 \times 512,$$
$$\text{LMU block (input Conv + LMU update + output Conv):} \quad N_{\text{LMU}} = 512^2 + 512^2 + 1024 \times 512,$$
$$\text{Conv channel mixer (2 Conv1d layers):} \quad N_{\text{CM}} = 2 \times 512 \times 512,$$
$$\text{Fully connected layer (512} \to \text{20):} \quad N_{\text{FC}} = 512 \times 20.$$

For sparse layers, we use the simplified energy model

$$E_{\text{sparse}}(N) = S \, N \, E_{\text{AC}},$$

while the first convolutional layer uses dense MAC:

$$E_{\text{dense}}(N) = N \, E_{\text{MAC}}.$$

**MAC+AC energy of Spiking LMUFormer (1 LMU block).** Using input width $m$, hidden width $n$ and class count $K$, the per–timestep compute energy of the Spiking LMUFormer with a 5-layer convolutional patch embedding, one LMU block and a two-layer convolutional channel mixer can be expressed (ignoring spike-overhead and comparison costs) as

$$E_{\text{step}}^{(\text{MAC+AC})} = C_{\text{MAC}} E_{\text{MAC}} + C_{\text{AC}} E_{\text{AC}},$$

where

$$C_{\text{MAC}} = mn, \qquad C_{\text{AC}} = S(10n^2 + nK),$$

with $S$ denoting the average firing rate. Thus

$$E_{\text{step}}^{(\text{MAC+AC})} = mn \, E_{\text{MAC}} + S(10n^2 + nK) E_{\text{AC}}.$$

**Energy per component (MAC+AC only).** The energy of each module is then

$$E_{\text{conv0}} = N_{\text{conv0}} E_{\text{MAC}} = (700 \times 512) \, E_{\text{MAC}} \approx 1.65 \times 10^6 \text{ pJ},$$
$$E_{\text{patch}} = E_{\text{sparse}}(N_{\text{patch}}) = S \, N_{\text{patch}} E_{\text{AC}} \approx 1.42 \times 10^5 \text{ pJ},$$
$$E_{\text{LMU}} = E_{\text{sparse}}(N_{\text{LMU}}) = S \, N_{\text{LMU}} E_{\text{AC}} \approx 1.42 \times 10^5 \text{ pJ},$$
$$E_{\text{CM}} = E_{\text{sparse}}(N_{\text{CM}}) = S \, N_{\text{CM}} E_{\text{AC}} \approx 7.08 \times 10^4 \text{ pJ},$$
$$E_{\text{FC}} = E_{\text{sparse}}(N_{\text{FC}}) = S \, N_{\text{FC}} E_{\text{AC}} \approx 1.38 \times 10^3 \text{ pJ}.$$

**Total per–timestep energy (MAC+AC only).**  Summing all MAC and AC contributions yields

$$E_{\text{step}}^{(\text{MAC+AC})} = E_{\text{conv0}} + E_{\text{patch}} + E_{\text{LMU}} + E_{\text{CM}} + E_{\text{FC}}$$
$$\approx 2.00 \times 10^6 \text{ pJ} = 2004 \text{ nJ}.$$

### A.9 ENERGY COUNTING OF LRMM ON SHD

**Setup.**  Two LRMM layers, each with $n_h = 512$ neurons split evenly: $|N_I| = |N_M| = |N_C| = |N_O| = 128$. Layer–1 uses external input of width $m_1 = 700$ with firing rate $Fr_{\text{in}}^{(1)} = 0.114$. Layer–2 takes input from Layer–1's output sub-population, hence $m_2 = |N_O^{(1)}| = 128$ and $Fr_{\text{in}}^{(2)} = Fr_{N_O}^{(1)}$. A final linear readout maps $|N_O^{(2)}| = 128$ to 20 classes. Per-neuron temporal updates contribute $n_h E_{\text{MAC}}$ per step, and all event-triggered operations (input projections, output fan-out, and local-module interactions) are accounted as accumulation events with cost $E_{\text{AC}}$.

**Per-layer counting (using total $n$).**  Let $n = 512$ be the total number of hidden neurons per LRMM layer. For layer $\ell \in \{1, 2\}$ with input width $m_\ell$ and input firing $Fr_{\text{in}}^{(\ell)}$, the per-step energy is

$$E_{\text{step}}^{(\ell)} = E_A^{(\ell)} + E_B^{(\ell)} + E_C^{(\ell)} + E_D^{(\ell)} + E_E^{(\ell)}$$
$$= \frac{7}{4} n E_{\text{MAC}} + \left( \frac{3m_\ell n}{4} Fr_{\text{in}}^{(\ell)} + \frac{3n^2}{16} Fr_{N_O}^{(\ell)} + \frac{n}{4} (Fr_{N_I}^{(\ell)} + Fr_{N_M}^{(\ell)} + 2\, Fr_{N_C}^{(\ell)}) \right) E_{\text{AC}}. \tag{48}$$

**Firing rates (given).**

$$\text{Layer 1: } (Fr_{N_I}^{(1)}, Fr_{N_O}^{(1)}, Fr_{N_C}^{(1)}, Fr_{N_M}^{(1)}) = (0.168,\, 0.311,\, 0.366,\, 0.274),$$
$$\text{Layer 2: } (Fr_{N_I}^{(2)}, Fr_{N_O}^{(2)}, Fr_{N_C}^{(2)}, Fr_{N_M}^{(2)}) = (0.196,\, 0.269,\, 0.330,\, 0.197).$$

**Layer–1** ($m_1{=}700$, $Fr_{\text{in}}^{(1)}{=}0.114$).  Substituting into equation 48 gives

$$E_{\text{L1}} = 896\, E_{\text{MAC}} + 46079.744\, E_{\text{AC}}.$$

**Layer–2** ($m_2{=}128$, $Fr_{\text{in}}^{(2)}{=}Fr_{N_O}^{(1)}{=}0.311$).

$$E_{\text{L2}} = 896\, E_{\text{MAC}} + 28642.944\, E_{\text{AC}}.$$

**Final readout** ($|N_O^{(2)}|{=}128 \to 20$).  Under the same event-driven assumption, the linear readout cost per step is

$$E_{\text{readout}} = (128 \times 20 \times Fr_{N_O}^{(2)})\, E_{\text{AC}} = (2560 \times 0.269)\, E_{\text{AC}} = 688.64\, E_{\text{AC}}. \tag{49}$$

**Two-layer total per step (with readout).**  Summing the two LRMM layers and the readout yields

$$E_{\text{total/step}} = 1792\, E_{\text{MAC}} + 75411.328\, E_{\text{AC}}. \tag{50}$$

### A.10 ABLATION STUDY ON CURRENT COMBINATIONS.

Table 5 investigates how different combinations of currents affect performance on PS-MNIST. Using the full triplet $(I_I{+}I_F{+}I_O)$ yields the highest accuracy of 96.52%, while removing any single current consistently degrades performance: the best two-current variant $I_I{+}I_F$ drops to 94.45%, and $I_I{+}I_O$ and $I_F{+}I_O$ further decrease to 93.14% and 93.60%, respectively. When only a single current is retained, accuracy falls to around 90% for $I_I$ and $I_F$, and down to 84.91% for $I_O$ alone. These results indicate that the three currents contribute synergistically rather than redundantly, and that the input and forget currents ($I_I$ and $I_F$) are particularly crucial for maintaining high task performance, while $I_O$ alone is insufficient but still provides complementary gains when combined with the others.

Table 4: Total per-step energy on SHD (two hidden layers, $n$=512 each). Counts follow $E_{\text{total}} = \#\text{MAC} \cdot E_{\text{MAC}} + \#\text{AC} \cdot E_{\text{AC}}$ with $E_{\text{MAC}} = 4.6\,\text{pJ}$ and $E_{\text{AC}} = 0.9\,\text{pJ}$. LRMM total includes the $128{\to}20$ readout.

| Model (2 layers) | Theoretical per-step count | Measured energy (nJ) |
|---|---|---|
| LRMM (2 layers + out) | $1792\,E_{\text{MAC}} + 75{,}411.328\,E_{\text{AC}}$ | **76.11** |
| LIF (2 layers) | $1024\,E_{\text{MAC}} + 202{,}097.664\,E_{\text{AC}}$ | 186.5983 |
| TC-LIF (2 layers) | $2048\,E_{\text{MAC}} + 225{,}390.592\,E_{\text{AC}}$ | 212.2723 |
| LSTM (2 layers) | $4{,}596{,}736\,E_{\text{MAC}} + 0\,E_{\text{AC}}$ | 21,144.9856 |
| Spiking LMUFormer (1 layers) | $358{,}400\,E_{\text{MAC}} + 2{,}631{,}680\,E_{\text{AC}}$ | 2004 |

| Currents Combo | $I_I{+}I_F{+}I_O$ | $I_I{+}I_F$ | $I_I{+}I_O$ | $I_F{+}I_O$ | $I_I$ | $I_F$ | $I_O$ |
|---|---|---|---|---|---|---|---|
| Acc | 96.52 | 94.45 | 93.14 | 93.60 | 90.28 | 89.76 | 84.91 |

Table 5: Ablation of current combinations with test accuracy on PS-MNIST.

## A.11 ABLATION EXPERIMENTS ON PARAMETER SENSITIVITY.

For parameter sensitivity, we conduct a grid-search by varying the initialization of two key hyperparameters and measuring the resulting test accuracy on PS-MNIST. The first hyperparameter is the leak factor $LF = 1 - 1/\tau$, i.e., the per-timestep decay multiplier applied to the membrane potential. The second hyperparameter is the recurrent coupling strength between the memory and context populations: we initialize both $w_{M,C}$ and $w_{C,M}$ to the same value and sweep this shared weight.

As shown in Table 6, LRMM remains highly stable across a wide range of leak factors. Even with very strong leakage ($LF = 0.10$) or very slow decay ($LF = 0.99$), the model consistently achieves around 95–96.5% accuracy, with the best performance at $LF = 0.95$. Similarly, Table 7 shows that varying the initial recurrent weight between $N_M$ and $N_C$ from 0.10 to 1.00 leads to only modest accuracy fluctuations, with all settings reaching above 94.5% and a broad optimum around 0.5–0.8, suggesting that LRMM does not require delicate hyperparameter tuning and exhibits stable convergence behavior under a broad range of initializations.

## A.12 ABLATION STUDY ON MODULATION FUNCTION TRAINING

Table 8 evaluates the impact of the modulation function on PS-MNIST and SHD. With the modulation mechanism enabled, LRMM achieves 96.52% and 94.70% test accuracy on PS-MNIST and SHD, respectively. Removing the modulation function leads to a clear performance drop on both benchmarks, down to 92.81% on PS-MNIST and 93.17% on SHD.

The modulation function prevents the sigmoidal gating term from saturating by dynamically linearizing its effective range during training. Without this mechanism, replacing the gating nonlinearity with a static linear or clipped function makes the recurrent dynamics more likely to get stuck in poor local optima, resulting in less stable optimization and degraded accuracy. This ablation therefore confirms that the modulation function is an important component for learning reliable gating behaviour and approaching the best performance of LRMM on temporal benchmarks.

## A.13 FINE-GRAINED OPERATION-LEVEL ENERGY ANALYSIS

To better understand where energy is spent in LRMM, we break down its per-step consumption by operation type, as shown in Table 9. Most of the energy comes from spike-driven accumulations, especially from forward and recurrent connections, which together make up over 88% of the total. In comparison, the deterministic updates and the modulation-related computations, such as current scaling and local interactions, contribute much less. This shows that while LRMM introduces additional structure for memory, it does so with only a small increase in energy, making it a practical choice for efficient neuromorphic computing.

| $LF$ | 0.10 | 0.20 | 0.30 | 0.40 | 0.50 | 0.60 | 0.70 | 0.80 | 0.90 | 0.95 | 0.99 |
|------|------|------|------|------|------|------|------|------|------|------|------|
| Acc | 95.11 | 95.07 | 95.63 | 95.48 | 95.87 | 96.08 | 96.17 | 96.04 | 96.36 | 96.52 | 96.49 |

Table 6: Parameter Sensitivity to the Leak Factor $LF = 1 - 1/\tau$ Measured on PS-MNIST.

| $w$ | 0.10 | 0.20 | 0.30 | 0.40 | 0.50 | 0.60 | 0.70 | 0.80 | 0.90 | 1.00 |
|-----|------|------|------|------|------|------|------|------|------|------|
| Acc | 94.67 | 95.23 | 95.56 | 95.63 | 95.94 | 96.52 | 96.31 | 96.40 | 96.01 | 95.48 |

Table 7: Effect of the Initial $N_M - N_C$ Recurrent Weight on PS-MNIST Test Accuracy. Both $w_{M,C}$ and $w_{C,M}$ are initialized to the same value.

| Benchmarks | PS-MNIST | SHD |
|------------|----------|-----|
| w/ modulation | 96.52 | 94.70 |
| w/o modulation | 92.81 | 93.17 |

Table 8: Ablation study on the modulation function: test accuracy (%) on PS-MNIST and SHD with and without the modulation mechanism.

## A.14 ANALYSIS OF TRAINING TIME AND GPU MEMORY USAGE

Table 10 reports the training time and GPU memory usage per epoch for LRMM and RSNN across four representative benchmarks. We observe that LRMM requires approximately $3\times$ longer training time than RSNN, which can be attributed to the additional computation introduced by its local recurrent modules. Despite this, the GPU memory usage of LRMM remains comparable to that of RSNN. This is largely due to the sparse weight structure within LRMM, which reduces the memory overhead associated with synaptic storage and event buffering. Overall, the additional cost in runtime is moderate and comes with no significant memory penalty, supporting the practical deployability of LRMM in typical GPU-based training setups.

## A.15 PSEUDOCODE FOR LRMM FORWARD COMPUTATION

This pseudocode summarizes the core structure of LRMM's forward pass over time. It highlights the modular interaction between input, memory, context, and output subpopulations, as well as the current computation and event-driven LIF updates. While conceptually simple, the local recurrent loops enable rich temporal integration with minimal global recurrence.

| Related Eq(s) | Op. Type | Description | Energy (Table 3; nJ) |
|---|---|---|---|
| Eq. 2 | MAC | per-neuron deterministic update | 4.71 |
| Eq. 3 | AC | spike-driven forward connections | 41.95 |
| Eq. 3 | AC | spike-driven recurrent connections | 25.66 |
| Eq. 6, current terms in 7 9 | MAC | current scaling | 3.53 |
| Eqs. 8, event terms in 7 9 | AC | local LRMM module interactions | 0.26 |
| Total | | | 76.11 |

Table 9: Theoretical energy breakdown per operation type, grouped by their corresponding equations. MAC denotes multiply–accumulate, AC denotes accumulation of spike-driven events.

| Benchmark | LRMM | RSNN |
|---|---|---|
| *Training Time per Epoch* | | |
| SHD | 41s | 16s |
| PS-MNIST | 46m39s | 13m08s |
| SSC | 8m32s | 3m20s |
| Binary Adding Problem | 35s | 12s |
| *GPU Memory Usage (MB)* | | |
| SHD | 754 | 536 |
| PS-MNIST | 3144 | 1374 |
| SSC | 750 | 532 |
| Binary Adding Problem | 714 | 508 |

Table 10: Comparison of training time and GPU memory usage per epoch between LRMM and RSNN on four benchmarks.

---

**Algorithm 1:** One-Layer LRMM Forward Computation for Sequence $\{x_t\}_{t=1}^{T}$

---

**Input:** Input spike sequence $\{\text{input}_t\}_{t=1}^{T}$
**Output:** Output spike sequence $\{S_{N_O}[t]\}_{t=1}^{T}$

1 **Initialize:**
2 $U_{N_I}[0] = U_{N_M}[0] = U_{N_C}[0] = U_{N_O}[0] = 0$
3 $S_{N_I}[0] = S_{N_M}[0] = S_{N_C}[0] = S_{N_O}[0] = 0$
4 **for** $t = 1$ **to** $T$ **do**
5    **1. Compute currents from input and previous output:**
6    $\text{concat}[t] = [\text{input}[t]; S_{N_O}[t-1]]$
7    $I_I[t] = \Phi(W_I \cdot \text{concat}[t] + b_I)$
8    $I_F[t] = \Phi(W_F \cdot \text{concat}[t] + b_F)$
9    $I_O[t] = \Phi(W_O \cdot \text{concat}[t] + b_O)$
10    **2. Compute input current for each subpopulation:**
11    $I_{N_I}[t] = k_I \cdot I_I[t]$
12    $I_{N_M}[t] = w_{C,M} \cdot S_{N_C}[t-1] + k_F \cdot I_F[t]$
13    $I_{N_C}[t] = w_{I,C} \cdot S_{N_I}[t] + w_{M,C} \cdot S_{N_M}[t]$
14    $I_{N_O}[t] = w_{C,O} \cdot S_{N_C}[t] + k_O \cdot I_O[t]$
15    **3. Update LIF neurons and generate spikes:**
16    **foreach** $X \in \{N_I, N_M, N_C, N_O\}$ **do**
17       $U_X[t] = \alpha_X \cdot (U_X[t-1] - V_{\text{th},X} \cdot S_X[t-1]) + I_X[t]$
18       $S_X[t] = \mathbf{1}(U_X[t] > V_{\text{th},X})$
19    **4. Output spike:** $S_{N_O}[t]$

---