# OpenReview forum: "Unlocking Long-Term Dependencies in Spiking Neural Networks with a Recurrent LIF Memory Module"
_ICLR.cc/2026/Conference — ICLR 2026 Conference Desk Rejected Submission_

### Official Review · Reviewer_ncM5 · 2025-10-30

**Soundness:** 4
**Presentation:** 1
**Contribution:** 2
**Rating:** 4
**Confidence:** 4

**Summary:**

This paper proposes LRMM (Local Recurrent Memory Module), a novel architecture for spiking neural networks (SNNs) that enhances long-term dependency modeling without modifying the LIF neuron model itself. Instead of introducing adaptive thresholds or multiple compartments, the authors design a network-level microcircuit of four interacting LIF neurons (NI, NM, NC, NO), forming a local recurrent memory loop. They show that this loop structure improves gradient retention in backpropagation through time (BPTT), mitigating vanishing gradients and enhancing memory duration. Extensive experiments on PS-MNIST, SHD, SSC, and Binary Adding demonstrate state-of-the-art performance compared to LIF, ALIF, TC-LIF, and even LSTM, while preserving neuromorphic efficiency (up to 400× less energy).

**Strengths:**

1. The idea of improving long-term memory through network-level recurrence rather than neuron-level complexity is orthogonal to most prior works (ALIF, TC-LIF, RadLIF). The proposed LRMM microcircuit is interpretable, biologically plausible, and hardware-friendly.

2. Section 3.2 provides a rigorous derivation of the effective temporal gain in the NM–NC loop, offering a clear explanation for how gradient vanishing is mitigated (Eq. 18–21).This level of mathematical grounding is rare in SNN papers and adds significant credibility.

3. The method outperforms on all four standard benchmarks with fewer parameters (Table 1). Particularly, LRMM achieves 96.52% on PS-MNIST and 94.7% on SHD, competitive with or better than LSTM while maintaining 400× efficiency.

4. The authors systematically ablate the memory loop, gating separation, and recurrent feedback (Table 2), and visualize gate dynamics and gradient flow (Figs. 4–6), demonstrating both interpretability and robustness.

5. The detailed energy accounting (Section 4.4, Appendix A.7)

**Weaknesses:**

1. Some baseline results (e.g., ALIF, RadLIF) are taken from different sources rather than reimplemented under the same training regime.
While Appendix A.5 claims consistent settings, explicit mention of normalization and surrogate choices would strengthen fairness.

2. Figures 1–2 are dense; neuron connections could be better annotated to show causal order. Minor grammatical and typographical errors persist (e.g., “ensures and stable gradient flow”).

3. The mathematical formulas in the paper are not well formatted and are sometimes difficult to read intuitively. For instance, the use of italic vs. upright subscripts and inconsistent vector/matrix boldface could be improved according to the ICLR style guidelines. In particular, the subscripts in Equations (6–9) could be simplified to enhance readability.

4. The ablation study focuses on removing entire loops or gates, but does not analyze parameter sensitivity (e.g., effect of leak constant τ, weight magnitude w_NM,NC). This limits understanding of stability boundaries.

5. What would happen if the model were trained without the modulation function? This part of the method should also be included in the ablation study.

6. Since the paper claims to address the long-term dependency problem, it would be clearer to include the sequence length T of each dataset in Table 1. This would make the improvements on long-sequence tasks more intuitive.

7. The typesetting and layout could be polished to improve readability. Currently, the dense text and inconsistent formatting make it less smooth to read.

8. In Figure 5, the font size of the text at the bottom is too small to read comfortably. Please enlarge it for better visibility.

9. In Equations (6), (7), and (9), the variable $I$ is not binary, meaning that floating-point multiplications are still required. In the energy analysis, it would be helpful to provide a table showing the theoretical energy consumption for each operation (or each equation component) to allow clearer comparison.

10. Please report how much GPU memory and training time each task requires compared to a standard LIF-based RSNN baseline. This would help evaluate the actual computational overhead of LRMM.

11. You may consider adding pseudocode for the LRMM algorithm in the paper to make the implementation process clearer and easier to follow.

**Questions:**

See Weaknesses

---

> ### Author Response · Authors · 2025-11-22
>
> We appreciate the reviewer’s thoughtful suggestions. We have carefully addressed all comments and describe the changes in detail below. All changes are updated in the revised paper.
>
> ---
>
> ### W1: Some baseline results are directly cited rather than reimplemented.
>
>
> * We thank the reviewer for the comment. In the paper, we adopted the reported results from the original papers for comparison. We have also tried to reimplemented ALIF, TC-LIF, and RadLIF in our architecture with different surrogate, but they have shown lower performance than in their original paper. Therefore, we used their data as reference rather than the reimplemented results.
>
> ---
>
>
> ### Writing related: W2 & W3 & W6 & W7 & W8
> ### W2: Figures 1–2 are dense; better connection annotations and grammar cleanup are needed.
> ### W3: Math formatting is inconsistent.
> ### W6: Include sequence length T in Table 1 to support long-range dependency.
> ### W7: Improve typesetting and layout to enhance readability and visual clarity.
> ### W8: Text in Figure 5 is too small; increase font size for better visibility.
>
> * We sincerely apologize for our writing mistakes. We have made the following adjustments as reviewer pointed out. We have also extensively revised the paper to improve the writing.
>
> * 1. We improved the font in **Figure 1** to make it easier to read. In **Figure 2**, we used different colors and lines to represent different neurons and information flows to show a clearer causal sequence. And we corrected the grammatical and typographical issues in **line 496**.
> * 2. We improved the mathematical formatting in the paper according to the ICLR format guidelines, modifying the formatting of matrices and vectors, as shown in **Eq(3), line 147, 175. In equations (6-9)**, we simplified the scaling factors for currents and the weights of local connections to improve readability.
> * 3. For **Table 1 and Figure 3**, we added the sequence length T for each dataset to facilitate reading and understanding the capabilities for long sequence tasks.
> * 4. We reduced the layout density, giving all images more space around them, and standardized the image formatting to optimize readability.
> * 5. We enlarged the font in **Figure 5** for better readability.
>
>
> ---
>
> ### W4: The ablation focuses on structural components, but lacks parameter sensitivity analysis for stability.
>
> * Thanks for the comment. To analyze parameter sensitivity, we performed additional grid-search study by varying the initialization values of key hyperparameters and observing their effect on the final performance. On the PS-MNIST benchmark, we examined two key sensitivity dimensions, i.e., the leak factor $LF=(1-1/\tau)$ and the recurrent weights of the $N_M$ and $N_C$. The corresponding results are shown in the tables below.
>
> * Across a wide range of initialization values, the model consistently converges to high accuracy. This indicates that the LRMM microcircuit is stable with respect to initialization and can reliably adjust both the leak factor and the recurrent interaction weights during training. In other words, LRMM does not rely on carefully tuned initial values and robustly learns appropriate internal dynamics under diverse starting conditions.
>
> > Revision: The full sensitivity table and the analysis have been added to **Appendix A.11**.
>
> | $LF$  | 0.1   | 0.2   | 0.3   | 0.4   | 0.5   | 0.6   | 0.7   | 0.8   | 0.9   | 0.95  | 0.99  |
> |--------------|--------|--------|--------|--------|--------|--------|--------|--------|--------|--------|--------|
> |   PS-MNIST Acc    | 95.11% | 95.07% | 95.63% | 95.48% | 95.87% | 96.08% | 96.17% | 96.04% | 96.36% | 96.52% | 96.49% |
>
> | $w_{N_M,N_C}$,$w_{N_C,N_M}$ | 0.1   | 0.2   | 0.3   | 0.4   | 0.5   | 0.6   | 0.7   | 0.8   | 0.9   | 1.0  |
> |--------------|--------|--------|--------|--------|--------|--------|--------|--------|--------|--------|
> | PS-MNIST Acc  | 94.67% | 95.23% | 95.56% | 95.63% | 95.94% | 96.52% | 96.31% | 96.40% | 96.01% | 95.48% |

---

> ### Author Response · Authors · 2025-11-22
>
> ---
>
>
> ### W5: What would happen if the model were trained without the modulation function? This part of the method should also be included in the ablation study.
>
> * Thank you for the question. We performed an ablation study to evaluate the role of the modulation function during training. The results are summarized in the table below.
>
> |  | PS-MNIST|SHD|
> | ----------------------- | -- | -- |
> | w modulation function | 96.52%|94.70%
> |w/o modulation function | 92.81%|93.17% |
>
> * The modulation function prevents the nonlinear sigmoid term from saturating by dynamically linearizing its effective range during training. Without this mechanism, replacing the gating nonlinearity with a static linear or clipped function makes the recurrent dynamics prone to getting trapped in poor local optima. This leads to unstable optimization and significantly degraded performance. The ablation confirms that the modulation function is essential for learning reliable gating behavior and achieving the full performance of LRMM.
>
> > Revison: The complete table and the analysis have been added to Appendix A.12.
> ---
>
> ### W9: $I$ is not binary in Eqs. (6), (7), and (9). And a table of per-operation energy cost would help clarify the analysis.
>
> * Thanks for the question. In the original manuscript, we followed the energy calculation methodology of the TC-LIF model [1], specifically Eqs. (10) and (11). Although these include internal membrane–weight multiplications, the TC-LIF paper does not count them as additional MAC operations. As a result, TC-LIF reports one MAC per neuron per timestep in its energy model.
>
> * Following the reviewer’s suggestion, we recompute the energy model by counting all multiplications in Eqs. (6)–(9) as MAC operations, providing a more conservative upper-bound estimate.
>
> * To further illustrate the robustness of LRMM’s energy advantage, we also compute an expanded version of the energy model that includes every operation that could reasonablely be regarded as an extra costs. Even under this more inclusive accounting scheme, LRMM still exhibits significantly lower energy consumption than other spiking neuron models and LSTM architectures. We will update the revised paper to adopt this form of energy accounting.
>
> * The table summarizing the theoretical energy consumption of each operation is provided below.
>
>
>  Related Eq(s) | Op. Type| Description  | Energy in Table 3 |
> ---------|------------|---------------|----------------|
>  Eq. (2) | MAC | per-neuron deterministic update (leak, reset) | 4.71nJ |
>  Eq.(3) | AC | spike-driven forward connections | 41.95nJ |
> Eq.(3) | AC | spike-driven recurrent connections | 25.66nJ |
> Eq. (6), current terms from Eqs. (7),(9) | MAC | current scaling | 3.53nJ |
>  Eq.(8), event terms from Eqs. (7),(9) | AC | local LRMM module interactions | 0.26nJ |
> | Total |  |    | 76.11nJ |
>
> > Revision: The whole analysis and this table have been added to **Appendix A.13** in the revised paper.
>
> [1]Zhang, Shimin, et al. "Tc-lif: A two-compartment spiking neuron model for long-term sequential modelling." Proceedings of the AAAI conference on artificial intelligence. Vol. 38. No. 15. 2024.
>
> ---
>
> ### W10: Report GPU use and training time to compare LRMM with standard RSNNs.
>
> * We provide a table comparing LRMM and RSNN in terms of training time and GPU memory usage.
>
> |         | LRMM(SHD) | RSNN(SHD) |LRMM(PS-MNIST)|RSNN(PS-MNIST)|LRMM(SSC)|RSNN(SSC)|LRMM(Binary Adding problem)|RSNN(Binary Adding problem)|
> | ---------------|-------- | ------------  | -----------|-------------|------------|-----------|-----|------- |
> | Time per epoch    | 41s |16s   | 46mins 39s  | 13mins 8s|8mins 32s|3mins 20s|35s|12s
> | GPU memory  | 754M | 536M |3144M| 1374M| 750M |532M|714M|508M
>
> > Revision: The table and the analysis have been added to **Appendix A.14**.

---

> ### Author Response · Authors · 2025-11-22
>
> ### W11: Adding pseudocode to clarify LRMM’s implementation.
>
> * We thank the reviewer for the constructive suggestion. We agree that including pseudocode makes the implementation of LRMM clearer and easier to follow. We provide the pseudocode below.
>
> #### Algorithm: One-Layer LRMM Forward Computation for Sequence
>
> Input: Spike sequence $\{\text{input}[t]\}_{t=1}^{T}$
>
> Output: Output spike sequence ${S_{N_O}[t]}_{t=1}^{T}$
>
> Initialize:
> $U_{N_I}[0] = U_{N_M}[0] = U_{N_C}[0] = U_{N_O}[0] = 0$,
> $S_{N_I}[0] = S_{N_M}[0] = S_{N_C}[0] = S_{N_O}[0] = 0$
>
> for $t = 1$ to $T$ do
>
> 1. Compute currents from input and previous output:
>
>     $\text{concat}[t] = [\text{input}[t]; S_{N_O}[t-1]]$,
>
>     $I_I[t] = \Phi\big(W_I·\text{concat}[t] + b_I\big)$,
>
>     $I_F[t] = \Phi\big(W_F·\text{concat}[t] + b_F\big)$,
>
>     $I_O[t] = \Phi\big(W_O·\text{concat}[t] + b_O\big)$,
>
>
> 2. Current into each LIF neuron in every LRMM module:
>
>     $I_{N_I}[t] = k_{I} \cdot I_I[t]$,
>
>     $I_{N_M}[t] = w_{C,M} \cdot S_{N_C}[t-1] + k_{F} \cdot I_F[t]$,
>
>     $I_{N_C}[t] = w_{I,C} \cdot S_{N_I}[t] + w_{M,C} \cdot S_{N_M}[t]$,
>
>     $I_{N_O}[t] = w_{C,O} \cdot S_{N_C}[t] + k_{O} \cdot I_O[t]$
>
> 3. LIF membrane updates and spike generation for each neuron:
>
>     for any $X \in {N_I, N_M, N_C, N_O}$:
>
>     $U_X[t] = \alpha_X \big(U_X[t-1] - V_{th,X} S_X[t-1]\big) + I_X[t]$,
>
>     $S_X[t] = \mathbf{1}\big(U_X[t] > V_{th,X}\big)$
>
> 4. Record output spike:
>
>     $S_{N_O}[t]$ is the module output at time step $t$.
>
> end for
>
> > Revision: We have added an algorithm summarizing one forward pass of LRMM in **Appendix A.15**.

---

### Official Review · Reviewer_AM94 · 2025-11-01

**Soundness:** 3
**Presentation:** 3
**Contribution:** 2
**Rating:** 2
**Confidence:** 4

**Summary:**

This paper proposes a novel recurrent spiking neural network that performs well on several long-sequence tasks. It appears to be very efficient in parameter use and can use LIF and more advanced neuronal models. The paper includes a detailed analysis of the stability when training with BPTT.  The paper does much better than several reported competitors.

**Strengths:**

The description of the algorithm, stability analysis, and experimentation appear quite strong.

**Weaknesses:**

There are a couple of points in the presentation that are unclear.

1) The paper states "However, the gating mechanism is not natively supported by most neuromorphic processors, hindering their efficient implementation in SNNs."

I think it would be useful to further explain what specific aspect of the gating mechanism is not supported by hardware accelerators like Loihi 2.

2)  Alternatively,RSNNs suitable for long-sequence tasks may be obtained through complex network structure designs [ref],

"ref" is undefined.

3) The paper misses what I think is a key reference and competitor to this work.

   Z. Liu et al. "LMUFORMER: LOW COMPLEXITY YET POWERFUL SPIKING MODEL WITH LEGENDRE MEMORY UNITS", ICLR 2024. See https://arxiv.org/pdf/2402.04882.

This paper appears to outperform your work on PS-MNIST and does also very well on related speech tasks. It may, however, have many more parameters and thus a detailed comparison of the pros and cons of these different approaches seems quite important.

**Questions:**

1. How does this work compare to Z. Liu et al's, LMUFormer ICLR 2024 paper in terms of novelty and complexity. Both appear to propose recurrent spiking model architectures and both do well on long-range tasks yet the approaches seem somewhat different.  If this question gets properly addressed, I think this paper would be a nice additional contribution and I would like to re-adjust my score.

---

> ### Author Response · Authors · 2025-11-22
>
> We are grateful to the reviewers for the insightful comments. These comments have helped improve the quality of the manuscript. Our detailed responses to each point are as follows.
>
> ---
>
> ### W1: The paper mentions that gating is not natively supported by neuromorphic hardware, but it would be helpful to clarify which aspect of the gating mechanism is incompatible with platforms like Loihi 2.
>
> * Thanks for the great suggestions. Further explanations have been added to the paper in **Section 1, line 58-64** as follows:
>
>
> * "since many neuromorphic processors, such as Loihi 2(Abreu et al., 2025), rely on specialized arithmetic logic units (ALUs) to emulate neuron operations which only support low-precision, element-wise operations such as integer and fixed-point addition, multiplication, and bitwise shifts, which imposes inherent limitations when approximating nonlinear functions such as tanh, sigmoid or exponential functions. Directly performing these non-native operations using these ALUs might suffer significant accuracy loss and require support from external high precision units.  "
>
> > Revision:  Further explanations have been added to the paper in **Section 1, line 58-64**.
>
> ---
>
> ### W2: An undefined ref.
>
> * We sincerely apologize for the mistake. The missing reference links have now been added to the revised manuscript in **line 65**. We have also extensively polished the paper to improve readability.
>
> ---
>
> ### W3&Q1: The paper seems to miss an important related work: LMUFormer (Z. Liu et al., ICLR 2024), which achieves strong results on PS-MNIST and speech tasks. Since both models are recurrent SNNs designed for long-range tasks, it would be helpful to see a comparison in terms of novelty, complexity, and trade-offs.
>
> * We thank the reviewer for pointing out LMUFormer, which is indeed a related and important work. We have referenced it and included comparisons in the manuscript.
>
> * Comparison of LRMM and spiking LMUFormer:
> Spiking LMUFormer is a deep and complex model that integrates 7 Conv1d layers, 7 BatchNorm1d layers, and 7 SNN layers, an additional Conv channel-mixing block and a state-space memory module. Recurrent connections are only a small part of the Spiking LMUFormer. LRMM, on the other hand, is a more standard RSNN architecture with novel hierarchical global and local recurrent connections, it does not require convolution, BN or channel-mixing layers, making its structure simple and compact.
>
> * The attention-based approach of Spiking LMUFormer provides strong capability for long sequence modeling, which is evident, but it comes at the cost of a large number of parameters and dense computations. In contrast, LRMM achieves comparable performance on the PS-MNIST task while using only a fraction of the parameters, as shown in the table below, and consequently offering significantly better energy efficiency. It suggests that LRMM is a more sustainable architecture for edge applications, where computing resources and energy consumption are key limiting factors for large models. The use of only vanilla neurons and standard SNN operations also makes LRMM very compatible with different neuromorphic chips.
>
> > Revision: The comparison and discussions have been added to **Section 4.2 line 354-360, Section 4.4 line 487, Tabel 3 line 437 and Table 1 line 276, 284, 290** in the revised paper.
>
>
> |  | LRMM  | LRMM-ALIF | Spiking LMUFormer|
> | ----------------------- | -- | -- | -|
> | PS-MNIST Acc | 96.52%|97.39%|97.92%|
> | Number of model parameters on PS-MNIST | 0.15M|0.15M |1.61M
> | SHD Acc | 94.70%|95.32%|91.04%|
> | Number of model parameters on SHD | 0.42M|0.42M |2.12M
> |Energy per step per sample on SHD|76.11nJ|78.94nJ|2004nJ

---

> > ### Comment · Reviewer_AM94 · 2025-11-26
> > **Response to rebuttal**
> >
> > Thank you for addressing my concerns and adding the comparison to LMUFormer. I am increasing mys core.

---

> > > ### Author Response · Authors · 2025-11-26
> > >
> > > We sincerely appreciate your thoughtful assessment and are grateful for the improved score. We are fully committed to addressing any further questions.

---

### Official Review · Reviewer_syKC · 2025-11-01

**Soundness:** 2
**Presentation:** 3
**Contribution:** 3
**Rating:** 6
**Confidence:** 3

**Summary:**

The paper introduces a Recurrent LIF Memory Module (LRMM) as a network-level solution to enhance long-term dependencies in recurrent spiking neural networks (RSNNs) while preserving the efficiency of LIF neurons. Unlike prior approaches that modify individual neuron dynamics, LRMM use 4 LIF node to build a Network-Level Micro-circuit, and control them through a key local memory loop and three dynamically modulated currents.

**Strengths:**

1. It demonstrates that we need not design more complex, computationally expensive neurons to achieve long-term memory. Instead, it shows that through delicate network topology design, the simplest components (vanilla LIF) can achieve equivalent or even superior performance.

2. The paper clearly proves mathematically why its $N_M-N_C$ memory loop solves the vanishing gradient problem. This "effective temporal gain" is a clear and analyzable mechanism.

**Weaknesses:**

1. Although using LIF is advantageous, the complex local connections (fully connected microcircuits of four neurons) and global connections ($N_O$ to feedback to the gated FC) in LRMM may introduce significant routing overhead on physical chips, whereas many neuromorphic hardware implementations favor sparse or simpler local connections.

2. The Independent Contribution of Gates: The experiment combined the three gates ($I_I, I_F, I_O$) for ablation study, but this does not reveal which gate is most critical. A more refined ablation experiment should test the impact of removing (e.g, $I_F$ (forgetting current) or $I_O$ (output current) individually). This is crucial for understanding whether LRMM truly learned LSTM-like “forgetting” and “output” control mechanisms.

**Questions:**

See the weakness.

---

> ### Author Response · Authors · 2025-11-22
>
> We sincerely thank the reviewer for the constructive feedback. We have revised the manuscript and provide detailed responses below. All changes are updated in the revised paper.
>
> ---
>
> ### W1: LRMM’s dense local and global connections may lead to high routing costs, making it less friendly to neuromorphic hardware that prefers sparse wiring.
>
> - We agree that hardware friendliness should also consider other factors other than neuron simplicity. We think LRMM is still efficient in hardware deployment.
>
> - **Routing overhead**: when the newtork is placed in mutliple cores and dense connections across cores generate significant routing overhead. LRMM is a compact model that could be placed into a local neuromorphic core. For example, for PS-MNIST task, LRMM uses 1,024 neurons with 150,912 weights to achieve comparable performance to Spiking LMUFormer(Liu et al., 2024), an attention-based network with 1.5M weights. The compactness of LRMM reduces its core-to-core routing overhead.
>
> - **Processing overhead**: recurrent connections processed inside the core generate local data shuttling between processing units (e.g. ALU) and SRAM units, which depends on spike activities. Since the recurrent spike activities of LRMM are still sparse, the processing overhead for recurrent connections will be significantly less than that of the fully connections.
>
> - We thank the reviewer for raising this point. It is valuable to explore whether these local fully connections can be reduced by methods such as purning or local receptive field. We will continue to improve it in future work.
>
> ---
>
> ### W2: Ablation study on independent contributions of gating currents.
>
> * Thanks for these valuable suggestions. We conducted additional ablation studies on different combinations of gating currents, and the results are summarized in the table below. The experiments indicate that each current contributes complementary benefits, and removing any component leads to a measurable degradation in performance.
>
>
> | Currents Combo | $I_I+I_F+I_O$ | $I_I+I_F$ | $I_I+I_O$ | $I_F+I_O$ | $I_I$ | $I_F$ | $I_O$ |
> |------|--------|-----------|-----------|-----------|-------|-------|-------|
> | PS-MNIST ACC%    | 96.52         | 94.45     | 93.14     | 93.60     | 90.28 | 89.76 | 84.91 |
>
> > Revision: The full ablation table and analysis have been added to **Appendix A.10**.

---

> > ### Comment · Reviewer_syKC · 2025-11-27
> >
> > Thank you for the response. I would like to keep my score as it is since it is already positive, and I will continue to follow the discussion with the other reviewers.

---

> > > ### Author Response · Authors · 2025-11-27
> > >
> > > We appreciate your positive assessment and your continued engagement in the discussion. We are available to provide any additional clarification if needed.

---

### Official Review · Reviewer_YcJU · 2025-11-02

**Soundness:** 4
**Presentation:** 2
**Contribution:** 3
**Rating:** 4
**Confidence:** 4

**Summary:**

This paper proposes a Local Recurrent Memory Module (LRMM) based on vanilla Leaky Integrate-and-Fire (LIF) neurons to address long-term dependency challenges in recurrent spiking neural networks (RSNNs), which enhances long-term information storage via dynamic interactions of LIF neurons and global recurrent connections.

**Strengths:**

1. The proposed Local Recurrent Memory Module (LRMM) achieves both superior performance and high energy efficiency.

2. LRMM mitigates the vanishing gradient problem in backpropagation through time (BPTT) via its memory loop design and maintains hardware compatibility by relying on vanilla LIF neurons instead of complex neuron-level modifications.

**Weaknesses:**

1.  The model’s effectiveness is currently only validated on specific long-sequence benchmarks and some simple tasks,  and it lacks performance in larger-scale network structures on more complex tasks.

2. The presentation of this paper needs improvement; first, the citation format is not user-friendly for reading, and second, there are many minor errors (line 60 ref,  the figures are not cited in the main text).

3. I did not grasp the specific modifications that the authors proposed for the LRMM relative to Recurrent Spiking Neural Networks (RSNNs); my understanding is that a new architecture was proposed rather than modifications at the neuron level, yet the experiments primarily compare different types of neurons.

**Questions:**

Table 3 only accounts for the power consumption of synapses, while the power consumption of neurons should also be included.

---

> ### Author Response · Authors · 2025-11-22
>
> We appreciate the constructive comments from the reviewers. We have made corresponding revisions to improve the clarity and readability of the paper with additional data and analysis. Detailed responses are shown below.
>
> ---
> ### W1: The model’s effectiveness is currently only validated on specific long-sequence benchmarks and some simple tasks, and it lacks performance in larger-scale network structures on more complex tasks.
>
> - We agree that evaluating LRMM on more complex tasks would further demonstrate its robustness and generality. These tasks require substantial computing resources and long training period. We are actively working on new experiments to complement existing results.
>
> - We would also like to highlight the contribution of LRMM for its compactness and high efficiency compared to more complex networks. In addition to the demonstrated advantages over LSTM, ALIF, and TC-LIF, we further compared our method with Spiking LMUFormer(Liu et al., 2024), an attention-based network with much more complex structure which is widely regarded as one of the strongest SNN architectures for long-sequence tasks. The LRMM achieved comparable or better accuracy than Spiking LMUFormer on PS-MNIST, SHD and SSC datasets, while using only a fraction of the parameters (e.g. 0.15M vs 1.61M for PS-MNIST) and 26 $\times$ improvement in efficiency (76.11 nJ vs 2004 nJ to process one time step data in SHD). The balance between performance and energy efficiency over these medium level tasks makes LRMM a compelling choice for use on edge devices, and the effiicency will also benefit the fast inference of LRMM on larger and more complex tasks.
>
> > Revision: The comparison with Spiking LMUFormer and discussions of LRMM on medium long sequence tasks have been added to **Section 4.2 line 354-360, Section 4.4 line 487, Tabel 3 line 437 and Table 1 line 276, 284, 290** in the revised paper.
>
>
> ---
>
> ### W2: The paper’s formatting is unclear, with citation issues and missing figure references.
>
> - We sincerely apologize for the writing issues. We have revised the citation format to a more reader-friendly style, corrected the missing reference in line 64, and ensured that every figure is properly cited. We have also polished the writing for better clarity and readability.
>
> ---
>
> ### W3: It’s unclear what exactly LRMM changes compared to a regular RSNN. It looks like a new architecture, but the experiments mostly compare neurons.
>
> - Thank you for the comment and we apologize for the confusion. The reviewer is correct that our work is a new architecture rather than neuron level modifications. Compared with a standard RSNN, LRMM introduced a hierarchical global and local recurrent architecture for better long sequence modeling capability.
>
> - A motivation of our work is to build a neuromorphic hardware-friendly RSNNs that avoid non-spiking operations such as gating, softmax and batch norm. Several architecture studies for long sequence modeling have been proposed, such as spiking state-space models [1], convolutional SNNs [2], and transformer-like SNNs [3]. However, these architectures incorporated signficant amount of non-spiking operations. As a result, we kept them away from our comparisons in the original paper.
>
> > Revision: Following the reviewer's comment, we have revised the paper to include discussion of architecture design in **Section 1, line 69-70** and expanded experimental comparison to other architectures in **Section4.2**.
>
> [1] Shen, Shuaijie, et al. "Spikingssms: Learning long sequences with sparse and parallel spiking state space models." Proceedings of the AAAI Conference on Artificial Intelligence. Vol. 39. No. 19. 2025.
>
> [2] Xing, Yannan, Gaetano Di Caterina, and John Soraghan. "A new spiking convolutional recurrent neural network (SCRNN) with applications to event-based hand gesture recognition." Frontiers in neuroscience 14 (2020): 590164.
>
> [3] Liu, Zeyu, et al. "LMUFormer: Low Complexity Yet Powerful Spiking Model With Legendre Memory Units." The Twelfth International Conference on Learning Representations.
>
> ---
> ### Q1: Table 3 only accounts for the power consumption of synapses, while the power consumption of neurons should also be included.
>
> - The power consumption values shown in Table 3 are derived from the calculations in Appendix A.7 to A.9. The layer-by-layer calculation of power consumption is discussed in Appendix A.8 and A.9, which used the results of each layer discussed in A.7. In A.7, the power consumption of neurons is determined by the full cost of neuron state updates, as the term $E_A = nE_{MAC}$ (30). The overall power consumption shown in A.8 therefore already contained the power consumption of neurons.

---

### Author Response · Authors · 2025-11-22

Dear Reviewers,

Thank you very much for your insightful comments and suggestions that have helped us to substantially improve our manuscript. Below is a summary of the key changes made:

----
### 1. Highlighted Contribution

1. Clarified the key contribution of LRMM as a compact and neuromorphic hardware-friendly architecture for long sequence tasks, which shows **a balance of performance and efficiency**. The discussion has been added to **Section 4.2**. The suitability for neuromorphic hardware has been added to **Section 1**.

----
### 2. Improved Article Structure and Readability

1. Revised the information flow in **Figure 2**, using neurons of different colors and different lines to clearly illustrate the propagation of temporal and spatial information.
2. Enlarged the font in **Figure 1** and **Figure 5** for easier reading.
3. Simplified the subscripts in **Eqs. (6)-(9)** for easier reading, and changed the formulas to the ICLR recommended format in **Section 3**.
4. Fixed the grammar error in **Section 5** and an undefined citation in **Section 1**, and made the citation format more reader-friendly in the whole paper.
5. Supplemented the pseudocode for LRMM in **Appendix A.15**.
6. Extensively polished the writing of the paper.

----
### 3. Expanded experiments

1. Added a new baseline Spiking LMUFormer and compared it with our LRMM in **Table 1, 3 Section 4.2, 4.4**, demonstrating the trade-off of LRMM in capability and parameter and power consumption.
2. Supplemented several ablation experiments, including experiments of different gating current combinations in **Appendix A.10**, experiments of the stability of our initial parameter boundaries in **Appendix A.11** and experiments of the modulation function in **Appendix A.12**.
3. Provided energy consumption analysis of different operations to facilitate understanding of the power consumption of each step in **Appendix A.13** and updated a new wider-bounded power consumption calculation in **Table 3, Section 4.4 and Appendix A.7-9**.
4. Supplemented the time and GPU memory usage of LRMM compared to RSNN, in **Appendix A.14**.

----
We believe that we have substantially improved the paper. The point-to-point responses are added below. We would like to thank you again for your valuable time and constructive comments. We are open to any further suggestions.

Sincerely,

The Authors

---

### Comment · Area_Chair_ZK57 · 2025-11-24
**Please respond to the authors' rebuttal**

Dear reviewers,

The authors have now posted their rebuttal. Please review it and submit your responses as soon as possible so that they still have adequate time to address any remaining questions or concerns. Please note that the discussion period between authors and reviewers will close on December 3, 11:59 PM AOE, after which no further comments can be exchanged.

Best, Your AC

---

### Author Response · Authors · 2025-12-03

Dear AC,

Thank you for your time and effort in evaluating our paper. We appreciate your extensive work involved in this unique review process. We have made significant improvements during the rebuttal stage. Below is a brief summary of the key updates for your consideration.

1. **Summary of Strengths (acknowledged by our reviewers)**

    As reviewers pointed out in their comments, we proposed a novel RSNN architecture with hierarchical recurrent connections for long sequence modeling, whose microcircuit is **"interpretable, biologically plausible, and hardware-friendly"**, resulting in **"both superior performance and high energy efficiency"**, **"demonstrating that we need not design more complex, computationally expensive neurons to achieve long term memory"** but only with **"simplest components (vanilla LIF)"**. We also provided **"rigorous derivation"** explaining how the memory loop **"mitigates the vanishing gradient problem"** with **"quite strong"** algorithm, stability analysis, and experimentation.

    During the rebuttal stage, we provided further comparison to a more complex architecture such LMUFormer (attention-based SNN with SOTA performance), and added extensive ablation studies to further improve our work.

2. **Highlighted responses to Reviewers**

* **Reviewer YcJU**: 4 → 4 (did not respond)

    **Q**: Validation of LRMM in larger-scale network structures on more complex tasks.

    **A**: We clarified that the strength of our work lies in its efficiency, achieving comparable performance to larger-scale network structures (LMUFormer) with far fewer parameters (0.15M vs. 1.61M) and substantially lower energy (76.11 nJ vs. 2004 nJ).

    **Q**: Comparison at architecture level rather than neuron level.

    **A**: Added comparison to LMUFormer.

* **Reviewer syKC**: 6 → 6 (responded on Nov. 27th, 10 hours before the identity leak)

    **Q**: Hardware compatibility regarding communication overhead

    **A**: We discussed overhead for hardware deployment on mainstream neuromorphic chip and verified that LRMM is more friendly to be deployed on neuromorphic chip.

    **Q**: Additional ablation study on independent contribution of gating current.

    **A**: Added additional ablations and showed that each gate is both effective and complementary.

* **Reviewer AM94**: 2 → 6 (responded on Nov. 26th, 2 days before the identity leak)

    **Q**: Lack of clear differentiation from LMUFormer.

    **A**: We clarified that the two architectures are largely different from **(1) network structure**: our LRMM is a 2-layer network with RSNN structure while LMUFormer is a 7-layer conv+SNN design, and **(2) efficiencies**: LRMM provides compact size (1/10) and high energy efficiency (26×) with comparable performance.

* **Reviewer ncM5**: 4 → 4 (did not respond)

    **Q**: Detailed ablation studies over parameter initialization sensitivity, modulation-function impact, fine-grained energy and resource-usage analysis.

    **A**: We have provided all the suggested ablations and revisions and strengthened our work with better clarity.

*  We further **confirm** that we have strictly followed the double-blind review process and **have not made any attempt** for collusion or manipulation of the peer review process.

Sincerely,

The Authors

---

### Note · Program_Chairs · 2026-01-17
**Submission Desk Rejected by Program Chairs**

The following references in this submission do not refer to real documents and/or have major errors in bibliographic information:

 Zhiheng Yin et al. Temporal coupling enables long-term memory in spiking neural networks. In Proceedings of the IEEE/CVF Conference on Computer Vision and Pattern Recognition, 2023.
Tobias Bohnstingl, Johannes Brandstetter, Guillaume Bellec, and Wolfgang Maass. Radlif: A spiking neuron model with learned radial dynamics for long-term memory. arXiv preprint arXiv:2203.10192, 2022.